# On-Policy Policy Gradient Reinforcement Learning Without On-Policy Sampling

**Nicholas E. Corrado**                                          *ncorrado@wisc.edu*
*Department of Computer Sciences*
*University of Wisconsin–Madison*

**Josiah P. Hanna**                                              *jphanna@cs.wisc.edu*
*Department of Computer Sciences*
*University of Wisconsin–Madison*

**Reviewed on OpenReview:** *https://openreview.net/forum?id=nCoyFp8uO1*

## Abstract

On-policy reinforcement learning (RL) algorithms are typically characterized as algorithms that perform policy updates using i.i.d. trajectories collected by the agent's current policy. However, after observing only a finite number of trajectories, such on-policy sampling may produce data that fails to match the expected on-policy data distribution. This *sampling error* leads to high-variance gradient estimates that yield data-inefficient on-policy learning. Recent work in the policy evaluation setting has shown that non-i.i.d., off-policy sampling can produce data with lower sampling error w.r.t. the expected on-policy distribution than on-policy sampling can produce (Zhong et al., 2022). Motivated by this observation, we introduce an adaptive, off-policy sampling method to reduce sampling error during on-policy policy gradient RL training. Our method, **P**roximal **R**obust **O**n-**P**olicy **S**ampling (PROPS), reduces sampling error by collecting data with a *behavior policy* that increases the probability of sampling actions that are under-sampled w.r.t. the current policy. We empirically evaluate PROPS on continuous-action MuJoCo benchmark tasks as well as discrete-action tasks and demonstrate that (1) PROPS decreases sampling error throughout training and (2) increases the data efficiency of on-policy policy gradient algorithms.

## 1 Introduction

One of the most widely used classes of reinforcement learning (RL) algorithms is the class of on-policy policy gradient algorithms. These algorithms use gradient ascent on the parameters of a parameterized policy to increase the probability of observed actions with high expected returns under the current policy. The gradient is commonly estimated using the Monte Carlo estimator, an average computed over i.i.d. trajectories sampled from the current policy. The Monte Carlo estimator is consistent and unbiased; as the number of sampled trajectories increases, the empirical distribution of data converges to the expected distribution under the current policy, and thus the estimated gradient converges to the true gradient. However, the expense of environment interaction forces us to work with finite sample sizes. Thus, the empirical distribution of data often differs from the desired on-policy data distribution, a mismatch we call *sampling error*. Sampling error causes inaccurate gradient estimates, resulting in high-variance policy updates, slower learning, and potentially convergence to suboptimal policies.

With i.i.d. on-policy sampling, the only way to reduce sampling error is to collect more data. Alternatively, we can reduce sampling error more efficiently using adaptive, *off-policy* sampling. To illustrate this approach, consider an MDP with two discrete actions A and B, and suppose the current policy $\pi$ places equal probability on both actions in some state $s$. When following $\pi$, after 10 visits to $s$, we will observe both actions 5 times in expectation. Now suppose that after the first 9 visits to $s$, we actually observe A 4 times and B 5 times.

Figure 1: An overview of PROPS. Rather than collecting data $\mathcal{D}$ via on-policy sampling from the agent's current policy $\pi_{\boldsymbol{\theta}}$, we collect data with a separate data collection policy $\pi_{\boldsymbol{\phi}}$ that we continually adapt to reduce sampling error in $\mathcal{D}$ with respect to the agent's current policy.

If we sample an action from $\pi$ upon our next visit to $s$, we may sample B, and our data will not match the expected on-policy distribution. Alternatively, if we select the under-sampled action A with probability 1, we will observe each action 5 times, making the aggregate data match the on-policy distribution even though this final action was sampled off-policy. The first scenario illustrates on-policy sampling with sampling error; the second scenario uses adaptive, off-policy sampling to reduce sampling error.

Recently, Zhong et al. (2022) introduced an adaptive, off-policy sampling method (ROS) that can produce data that more closely matches the on-policy distribution than data acquired through i.i.d. on-policy sampling. However, this work was limited to low-dimensional policy evaluation tasks. Moreover, ROS required large batches of data to reduce sampling error—approximately 5000 samples on tasks like CartPole-v1 (Brockman et al., 2016)—and struggled to reduce sampling error on tasks with continuous actions. To make adaptive sampling practical for data-efficient RL, we need methods that can reduce sampling error in high-dimensional continuous-action tasks while using the smaller batch sizes typically used in RL. These observations raise the following question: *Can reducing sampling error through adaptive sampling increase the data-efficiency of on-policy policy gradient methods?*

In this work, we address these challenges and show for the first time that on-policy policy gradient algorithms are more data-efficient learners when they use adaptive, off-policy sampling to reduce sampling error w.r.t. the on-policy distribution. Our method, **P**roximal **R**obust **O**n-**P**olicy **S**ampling (PROPS), adaptively corrects sampling error in previously collected data by increasing the probability of sampling actions that are under-sampled with respect to the current policy. Fig. 1 provides an overview of PROPS. We empirically evaluate PROPS on continuous-action MuJoCo benchmark tasks and show that (1) PROPS reduces sampling error throughout training and (2) increases the data efficiency of on-policy policy gradient algorithms. In summary, our contributions are:

1. We introduce PROPS, a scalable adaptive sampling algorithm for on-policy policy gradient learning that reduces sampling error w.r.t. the agent's current policy.

2. We demonstrate empirically that PROPS reduces sampling error more efficiently than on-policy sampling and ROS.

3. We demonstrate empirically that reducing sampling error via PROPS increases data efficiency in on-policy policy gradient RL.

## 2   Related Work

**Data Collection and Exploration.** Our work focuses on data collection in RL. In RL, data collection is often framed as an exploration problem, focusing on how an agent should explore its environment

to efficiently learn an optimal policy. Prior works have proposed exploration-promoting methods such as intrinsic motivation (Pathak et al., 2017; Sukhbaatar et al., 2018), count-based exploration (Tang et al., 2017; Ostrovski et al., 2017), and Thompson sampling (Osband et al., 2013; Sutton & Barto, 2018). Rather than adjusting the agent's sampling distribution to promote exploration, our objective is to learn from the on-policy data distribution; we use adaptive data collection to more efficiently obtain this data distribution.

**Adaptive Sampling for Sampling Error Reduction.** Prior works have used adaptive off-policy sampling to reduce sampling error in the policy evaluation subfield of RL. Most closely related is the work of Zhong et al. (2022), who first proposed that adaptive off-policy sampling could produce data that more closely matches the on-policy distribution than on-policy sampling could produce. Mukherjee et al. (2022) use a deterministic sampling rule to take actions in a particular proportion. Other bandit works use a non-adaptive exploration policy to collect additional data conditioned on previously collected data (Tucker & Joachims, 2022; Wan et al., 2022; Konyushova et al., 2021). Since these works only focus on policy evaluation, they do not have to contend with a changing on-policy distribution as our work does for the control setting.

**Importance Sampling for Sampling Error Reduction.** Several prior works propose importance sampling methods (Precup, 2000) to reduce sampling error without further data collection. In the RL setting, Hanna et al. (2021) showed that reweighting off-policy data according to an estimated behavior policy can correct sampling error and improve policy evaluation. Liu et al. (2024) compute a globally optimal behavior policy using an offline dataset. Papini et al. (2024) focused on finding a behavior policy that minimizes the variance of importance-weighted policy gradient estimates. Similar methods have been studied for temporal difference learning (Pavse et al., 2020) and policy evaluation in the bandit setting (Li et al., 2015; Narita et al., 2019). Conservative Data Sharing (Yu et al., 2021) reduces sampling error by selectively integrating offline data from multiple tasks. While these works reduce sampling error by reweighting existing data, our work focuses on whether we can reduce sampling error *during data collection*. Moreover, works like Liu et al. (2024) require a large amount of offline data (*e.g.*, 1000 trajectories) to accurately compute a behavior policy, but in online on-policy RL, agents typically collect only a few trajectories between policy updates.

**On-Policy Learning with Off-Policy Data.** As we will discuss in Section 5, the method we introduce permits data collected in one iteration of policy optimization to be re-used in future iterations rather than discarded as typically done by on-policy algorithms. Prior work has attempted to avoid discarding data by combining off-policy and on-policy updates with separate loss functions or by using alternative gradient estimates (Wang et al., 2016; Gu et al., 2016; 2017; Fakoor et al., 2020; O'Donoghue et al., 2016; Queeney et al., 2021). In contrast, our method modifies the sampling distribution at each iteration so that the entire data set of past and newly collected data matches the expected distribution under the current policy.

## 3 Preliminaries

### 3.1 Reinforcement Learning

We formalize the RL environment as an infinite horizon Markov decision process (MDP) (Puterman, 2014) $(\mathcal{S}, \mathcal{A}, p, r, d_0, \gamma)$ with state space $\mathcal{S}$, action space $\mathcal{A}$, transition dynamics $p : \mathcal{S} \times \mathcal{A} \times \mathcal{S} \rightarrow [0,1]$, reward function $r : \mathcal{S} \times \mathcal{A} \rightarrow \mathbb{R}$, initial state distribution $d_0$, and reward discount factor $\gamma \in [0,1)$. The state and action spaces may be discrete or continuous. We consider stochastic policies $\pi_{\boldsymbol{\theta}} : \mathcal{S} \times \mathcal{A} \rightarrow [0,1]$ parameterized by $\boldsymbol{\theta}$, and we write $\pi_{\boldsymbol{\theta}}(\boldsymbol{a}|\boldsymbol{s})$ to denote the probability of sampling action $\boldsymbol{a}$ in state $\boldsymbol{s}$ and $\pi_{\boldsymbol{\theta}}(\cdot|\boldsymbol{s})$ to denote the probability distribution over actions in state $\boldsymbol{s}$. We additionally let $d_{\pi_{\boldsymbol{\theta}}} : \mathcal{S} \times \mathcal{A} \rightarrow [0,1]$ denote the state-action visitation distribution, the distribution over state-action pairs induced by following $\pi_{\boldsymbol{\theta}}$. The RL objective is to find a policy that maximizes the expected sum of discounted rewards, defined as:

$$J(\boldsymbol{\theta}) = \mathbb{E}_{\tau \sim \pi_{\boldsymbol{\theta}}} \left[ \sum\nolimits_{t=0}^{\infty} \gamma^t r(\boldsymbol{s}_t, \boldsymbol{a}_t) \right]. \tag{1}$$

Throughout this paper, we refer to the policy used for data collection as the *behavior policy* and the policy trained to maximize its expected return as the *target policy*.

## 3.2 On-Policy Policy Gradient Algorithms

Policy gradient algorithms perform gradient ascent over policy parameters to maximize an agent's expected return $J(\boldsymbol{\theta})$. The gradient of $J(\boldsymbol{\theta})$ with respect to $\boldsymbol{\theta}$, or *policy gradient*, is often given as:

$$\nabla_{\boldsymbol{\theta}} J(\boldsymbol{\theta}) = \mathbb{E}_{\boldsymbol{s} \sim d_{\pi_{\boldsymbol{\theta}}}, \boldsymbol{a} \sim \pi_{\boldsymbol{\theta}}} \left[ A^{\pi_{\boldsymbol{\theta}}}(\boldsymbol{s}, \boldsymbol{a}) \nabla_{\boldsymbol{\theta}} \log \pi_{\boldsymbol{\theta}}(\boldsymbol{a}|\boldsymbol{s}) \right], \tag{2}$$

where $A^{\pi_{\boldsymbol{\theta}}}(\boldsymbol{s}, \boldsymbol{a})$ is the *advantage* of choosing action $\boldsymbol{a}$ in state $\boldsymbol{s}$ and following $\pi_{\boldsymbol{\theta}}$ thereafter.[1] In practice, the expectation in Eq. 2 is approximated with Monte Carlo samples collected from $\pi_{\boldsymbol{\theta}}$, and an estimate of $A^{\pi_{\boldsymbol{\theta}}}$ is used in place of the true advantages (Schulman et al., 2016). Since off-policy data will bias this gradient estimator, on-policy algorithms discard historic off-policy data after each policy update and collect new data with the updated policy.[2]

This foundational idea of policy learning via stochastic gradient ascent was first proposed by Williams (1992) under the name REINFORCE. Since then, a large body of research has focused on developing more scalable policy gradient methods (Kakade, 2001; Schulman et al., 2015; Mnih et al., 2016; Espeholt et al., 2018; Lillicrap et al., 2015; Haarnoja et al., 2018). Currently, the most successful variant of on-policy learning is proximal policy optimization (PPO) (Schulman et al., 2017), the algorithm of choice in several high-profile success stories (Berner et al., 2019; Akkaya et al., 2019; Vinyals et al., 2019). PPO maximizes the clipped surrogate objective

$$\begin{aligned}
\mathcal{L}_{\text{PPO}}(\boldsymbol{s}, \boldsymbol{a}, \boldsymbol{\theta}, \boldsymbol{\theta}_{\text{old}}) = \min(&g(\boldsymbol{s}, \boldsymbol{a}, \boldsymbol{\theta}, \boldsymbol{\theta}_{\text{old}}) A^{\pi_{\boldsymbol{\theta}_{\text{old}}}}(\boldsymbol{s}, \boldsymbol{a}), \\
&\texttt{clip}(g(\boldsymbol{s}, \boldsymbol{a}, \boldsymbol{\theta}, \boldsymbol{\theta}_{\text{old}}), 1 - \epsilon, 1 + \epsilon) A^{\pi_{\boldsymbol{\theta}_{\text{old}}}}(\boldsymbol{s}, \boldsymbol{a})),
\end{aligned} \tag{3}$$

where $\boldsymbol{\theta}_{\text{old}}$ denotes the policy parameters prior to the update, $g(\boldsymbol{s}, \boldsymbol{a}, \boldsymbol{\theta}, \boldsymbol{\theta}_{\text{old}})$ is the policy ratio $g(\boldsymbol{s}, \boldsymbol{a}, \boldsymbol{\theta}, \boldsymbol{\theta}_{\text{old}}) = \frac{\pi_{\boldsymbol{\theta}}(\boldsymbol{a}|\boldsymbol{s})}{\pi_{\boldsymbol{\theta}_{\text{old}}}(\boldsymbol{a}|\boldsymbol{s})}$, and the $\texttt{clip}$ function with hyperparameter $\epsilon$ clips $g(\boldsymbol{s}, \boldsymbol{a}, \boldsymbol{\theta}, \boldsymbol{\theta}_{\text{old}})$ to the interval $[1 - \epsilon, 1 + \epsilon]$. This objective disincentivizes large changes to $\pi_{\boldsymbol{\theta}}(\boldsymbol{a}|\boldsymbol{s})$. While other policy gradient algorithms perform a single gradient update per data sample to avoid destructively large policy updates, PPO's clipping mechanism permits multiple epochs of minibatch policy updates.

## 4 Correcting Sampling Error in Reinforcement Learning

In this section, we illustrate how sampling error can produce inaccurate policy gradient estimates and then describe how adaptive, off-policy sampling can reduce sampling error. For exposition, we assume finite state and action spaces. The policy gradient can then be written as:

$$\nabla_{\boldsymbol{\theta}} J(\boldsymbol{\theta}) = \sum_{(\boldsymbol{s}, \boldsymbol{a}) \in \mathcal{S} \times \mathcal{A}} d_{\pi_{\boldsymbol{\theta}}}(\boldsymbol{s}, \boldsymbol{a}) \left[ A^{\pi_{\boldsymbol{\theta}}}(\boldsymbol{s}, \boldsymbol{a}) \nabla_{\boldsymbol{\theta}} \log \pi_{\boldsymbol{\theta}}(\boldsymbol{a}|\boldsymbol{s}) \right]. \tag{4}$$

The policy gradient is thus a linear combination of the gradient for each $(\boldsymbol{s}, \boldsymbol{a})$ pair $\nabla_{\boldsymbol{\theta}} \log \pi_{\boldsymbol{\theta}}(\boldsymbol{a}|\boldsymbol{s})$ weighted by $d_{\pi_{\boldsymbol{\theta}}}(\boldsymbol{s}, \boldsymbol{a}) A^{\pi_{\boldsymbol{\theta}}}(\boldsymbol{s}, \boldsymbol{a})$. Let $\mathcal{D}$ be a dataset of trajectories. It is straightforward to show that the Monte Carlo estimate of the policy gradient can be written in a form similar to Eq. 4 except with the true state-action visitation distribution replaced with the empirical visitation distribution $d_{\mathcal{D}}(\boldsymbol{s}, \boldsymbol{a})$, denoting the fraction of times $(\boldsymbol{s}, \boldsymbol{a})$ appears in $\mathcal{D}$ (Hanna et al., 2021):

$$\nabla_{\boldsymbol{\theta}} \widehat{J}(\boldsymbol{\theta}) = \sum_{(\boldsymbol{s}, \boldsymbol{a}) \in \mathcal{S} \times \mathcal{A}} d_{\mathcal{D}}(\boldsymbol{s}, \boldsymbol{a}) \left[ A^{\pi_{\boldsymbol{\theta}}}(\boldsymbol{s}, \boldsymbol{a}) \nabla_{\boldsymbol{\theta}} \log \pi_{\boldsymbol{\theta}}(\boldsymbol{a}|\boldsymbol{s}) \right]. \tag{5}$$

By comparing Eq. 4 and Eq. 5, we can understand how sampling error affects the Monte Carlo policy gradient estimate. When $(\boldsymbol{s}, \boldsymbol{a})$ is over-sampled (*i.e.*, $d_{\mathcal{D}}(\boldsymbol{s}, \boldsymbol{a}) > d_{\pi_{\boldsymbol{\theta}}}(\boldsymbol{s}, \boldsymbol{a})$), then $\nabla_{\boldsymbol{\theta}} \log \pi_{\boldsymbol{\theta}}(\boldsymbol{a}|\boldsymbol{s})$ contributes more to the overall gradient than it should. Similarly, when $(\boldsymbol{s}, \boldsymbol{a})$ is under-sampled, $\nabla_{\boldsymbol{\theta}} \log \pi_{\boldsymbol{\theta}}(\boldsymbol{a}|\boldsymbol{s})$ contributes less

---

[1]We note that Eq. 2 is a biased estimator of the policy gradient; the expectation is taken over the undiscounted state distribution $d_{\pi_{\boldsymbol{\theta}}}$ rather than the discounted state distribution $d_{\pi_{\boldsymbol{\theta}}}^{\gamma}$, so Eq. 2 is not the gradient of the discounted objective Eq. 1 (Nota & Thomas, 2019). Nevertheless, since many popular policy gradient algorithms estimate the policy gradient using Eq. 2 (Schulman et al., 2017; Haarnoja et al., 2018; Mnih et al., 2016), we use $d_{\pi_{\boldsymbol{\theta}}}$ rather than $d_{\pi_{\boldsymbol{\theta}}}^{\gamma}$ in this work.

[2]In practice, policy gradient algorithms have other sources of gradient bias, such as using $\lambda$-returns in place of Monte Carlo returns (Sutton & Barto, 2018) and using $d_{\pi_{\boldsymbol{\theta}}}^{\gamma}$ in place of $d_{\pi_{\boldsymbol{\theta}}}$ (Nota & Thomas, 2019).

than it should. Below, we provide a concrete example illustrating how small amounts of sampling error can cause the wrong actions to be reinforced.

---

**Example:** Sampling error can cause incorrect policy updates.

Let $\pi_{\boldsymbol{\theta}}$ be a policy for an MDP with two discrete actions $\boldsymbol{a}_0$ and $\boldsymbol{a}_1$, and suppose that in a particular state $\boldsymbol{s}_0$, the advantage of each action w.r.t. $\pi_{\boldsymbol{\theta}}$ is $A^{\pi_{\boldsymbol{\theta}}}(\boldsymbol{s}_0, \boldsymbol{a}_0) = 20$ and $A^{\pi_{\boldsymbol{\theta}}}(\boldsymbol{s}_0, \boldsymbol{a}_1) = 15$. For simplicity, suppose the policy has a direct parameterization $\pi_{\boldsymbol{\theta}}(\boldsymbol{a}_0|\boldsymbol{s}) = \boldsymbol{\theta}_{\boldsymbol{s}}$, $\pi_{\boldsymbol{\theta}}(\boldsymbol{a}_1|\boldsymbol{s}) = 1 - \boldsymbol{\theta}_{\boldsymbol{s}}$ and places equal probability on both actions in $\boldsymbol{s}_0$ ($\boldsymbol{\theta}_{\boldsymbol{s}_0} = 0.5$). Then, we have $\nabla_{\boldsymbol{\theta}} \log \pi_{\boldsymbol{\theta}}(\boldsymbol{a}_0|\boldsymbol{s}_0) = -\nabla_{\boldsymbol{\theta}} \log \pi_{\boldsymbol{\theta}}(\boldsymbol{a}_1|\boldsymbol{s}_0)$ and $d_{\pi_{\boldsymbol{\theta}}}(\boldsymbol{s}_0, \boldsymbol{a}_0) = d_{\pi_{\boldsymbol{\theta}}}(\boldsymbol{s}_0, \boldsymbol{a}_1)$ so that the expected gradient increases the probability of sampling $\boldsymbol{a}_0$, the optimal action. With on-policy sampling, after 10 visits to $\boldsymbol{s}_0$, the agent will sample both actions 5 times in expectation. However, if the agent actually observes $\boldsymbol{a}_0$ 4 times and $\boldsymbol{a}_1$ 6 times, a Monte Carlo estimate of the policy gradient then yields

$$\frac{4}{10} \cdot 20 \cdot \nabla_{\boldsymbol{\theta}} \log \pi_{\boldsymbol{\theta}}(\boldsymbol{a}_0|\boldsymbol{s}_0) + \frac{6}{10} \cdot 15 \cdot \nabla_{\boldsymbol{\theta}} \log \pi_{\boldsymbol{\theta}}(\boldsymbol{a}_1|\boldsymbol{s}_0) = -\nabla_{\boldsymbol{\theta}} \log \pi_{\boldsymbol{\theta}}(\boldsymbol{a}_0|\boldsymbol{s}_0)$$

which *decreases* the probability of sampling the optimal $\boldsymbol{a}_0$ action.

---

Sampling error in on-policy sampling vanishes as the size of the batch of data used to estimate the gradient tends toward infinity. However, the preceding example suggests a simple strategy to eliminate sampling error with finite data: have the agent adapt its probability on the next action it takes based on the actions it has already sampled. Continuing with our example, suppose the agent has visited $\boldsymbol{s}_0$ 9 times and sampled $\boldsymbol{a}_0$ 4 times and $\boldsymbol{a}_1$ 5 times. With on-policy sampling, the agent may observe $\boldsymbol{a}_1$ again upon the next visit to $\boldsymbol{s}_0$. Alternatively, the agent could sample its next action from a distribution that puts probability 1 on $\boldsymbol{a}_0$ and consequently produce an aggregate batch of data that contains both actions in their expected frequency. While this adaptive method uses off-policy sampling, it produces data that exactly matches the on-policy distribution and thus produces a more accurate gradient.

This example suggests that we can heuristically reduce sampling error by selecting the most under-sampled action at a given state. Under a strong assumption that the MDP has a directed acyclic graph (DAG) structure, Zhong et al. (2022) proved that in a fixed-horizon MDP, this heuristic produces an empirical state-action distribution that converges to $d_{\pi_{\boldsymbol{\theta}}}(\boldsymbol{s}, \boldsymbol{a})$ and moreover converges at a faster rate than on-policy sampling. We remove this limiting DAG assumption with the following result:

**Proposition 1.** *Assume that data is collected with an adaptive behavior policy that always takes the most under-sampled action in each state $s$ w.r.t. $\pi$, i.e., $a \leftarrow \arg\max_{a'}(\pi(a'|s) - \pi_{\mathcal{D}}(a'|s))$, where $\pi_{\mathcal{D}}$ is the empirical policy after $m$ state-action pairs have been collected. Assume that $\mathcal{S}$ and $\mathcal{A}$ are finite and that the Markov chain induced by $\pi$ is irreducible. Then we have that the empirical state visitation distribution, $d_m$, converges to the state distribution of $\pi$, $d_\pi$, with probability 1:*

$$\forall s, \lim_{m \to \infty} d_m(s) = d_\pi(s).$$

We prove Proposition 1 in Appendix A. While adaptively sampling the most under-sampled action can reduce sampling error, this heuristic is difficult to implement in practice; in tasks with continuous states and actions, the $\arg\max$ in Proposition 1 often has no closed-form solution, and the empirical policy $\pi_{\mathcal{D}}$ can be expensive to compute at every timestep. Building upon the concepts discussed in this section, the following section presents a *scalable* adaptive sampling algorithm that reduces sampling error in on-policy policy gradient learning.

## 5 Proximal Robust On-Policy Sampling for Policy Gradient Algorithms

Our goal is to develop an adaptive, off-policy sampling algorithm that reduces sampling error in on-policy data collection for on-policy policy gradient algorithms. We outline a general framework for on-policy learning with an adaptive behavior policy in Algorithm 1. The behavior policy collects a batch of $m$ transitions, adds the batch to a data buffer $\mathcal{D}$, and then updates its weights such that the next batch it collects reduces

sampling error in $\mathcal{D}$ with respect to the target policy $\pi_{\boldsymbol{\theta}}$ (Lines 7-10). Every $n$ steps (with $n > m$), the agent updates its target policy with data from $\mathcal{D}$ (Line 11). We refer to $m$ and $n$ as the *behavior update frequency* and the *target batch size*, respectively.

The behavior policy must continually adjust action probabilities for new samples so that the aggregate data distribution of $\mathcal{D}$ matches the expected on-policy distribution of the current target policy (Line 10). A subtle implication of this adaptive sampling is that it can correct sampling error in *any* empirical data distribution—even one generated by a different policy. Thus, rather than discarding off-policy data from old policies as is commonly done in on-policy learning, we let the data buffer hold up to $b$ target batches ($bn$ transitions) and call $b$ the *buffer size*. If $b > 1$, then $\mathcal{D}$ will contain historic off-policy data used in previous target policy updates.[3] Implementing Line 10 is the core challenge we address in the remainder of this section.

---

**Algorithm 1** On-policy policy gradient algorithm with adaptive sampling

---
1: **Inputs**: Target batch size $n$, behavior update frequency $m$, buffer size $b$.
2: **Output:** Target policy parameters $\boldsymbol{\theta}$.
3: Initialize target policy parameters $\boldsymbol{\theta}$.
4: Initialize behavior policy parameters $\boldsymbol{\phi} \leftarrow \boldsymbol{\theta}$.
5: Initialize empty buffer $\mathcal{D}$ with capacity $bn$.
6: **for** target update $i = 1, 2, \ldots$ **do**
7:     **for** behavior update $j = 1, \ldots, \lfloor n/m \rfloor$ **do**
8:         Collect batch of data $\mathcal{B}$ by running $\pi_{\boldsymbol{\phi}}$.
9:         Append $\mathcal{B}$ to buffer $\mathcal{D}$.
10:         Update $\pi_{\boldsymbol{\phi}}$ with $\mathcal{D}$ using Algorithm 2.
11:     Update $\pi_{\boldsymbol{\theta}}$ with $\mathcal{D}$.
12: **return** $\boldsymbol{\theta}$

---

### 5.1 Robust On-Policy Sampling

To ensure that the empirical distribution of $\mathcal{D}$ matches the expected on-policy distribution, updates to $\pi_{\boldsymbol{\phi}}$ should attempt to increase the probability of actions which are currently under-sampled with respect to $\pi_{\boldsymbol{\theta}}$. Zhong et al. (2022) recently developed a simple method called Robust On-policy Sampling (ROS) for making such updates, which PROPS builds upon. In this section, we briefly review ROS and discuss its limitations.

Let $\pi_{\mathcal{D}}(\boldsymbol{a}|\boldsymbol{s})$ denote the empirical policy, the fraction of times action $\boldsymbol{a}$ is taken at state $\boldsymbol{s}$ in $\mathcal{D}$. If $\pi_{\mathcal{D}}(\boldsymbol{a}|\boldsymbol{s}) < \pi_{\boldsymbol{\theta}}(\boldsymbol{a}|\boldsymbol{s})$, then $\boldsymbol{a}$ appears fewer times at $\boldsymbol{s}$ than it would in expectation under $\pi_{\boldsymbol{\theta}}$ and is therefore *under-sampled*. Thus, $\pi_{\mathcal{D}}$ provides a way to identify actions whose probabilities should be increased to reduce sampling error. In the tabular setting, $\pi_{\mathcal{D}}$ can be computed exactly as the maximizer of the log-likelihood,

$$\mathcal{L}(\boldsymbol{\theta}') = \sum\nolimits_{(\boldsymbol{s},\boldsymbol{a}) \in \mathcal{D}} \log \pi_{\boldsymbol{\theta}'}(\boldsymbol{a} \mid \boldsymbol{s}), \tag{6}$$

*i.e.*, $\pi_{\mathcal{D}} = \arg\max_{\boldsymbol{\phi}} \mathcal{L}(\boldsymbol{\theta}')$. However, in tasks with high-dimensional continuous state and action spaces, computing $\pi_{\mathcal{D}}$ explicitly is generally intractable. The key insight of ROS is that we do not need to compute $\pi_{\mathcal{D}}$ explicitly; we only need a direction that tells us how to adjust action probabilities based on how the data in $\mathcal{D}$ differs from $\pi_{\boldsymbol{\theta}}$. The gradient of the log-likelihood at $\boldsymbol{\phi} = \boldsymbol{\theta}$, $\nabla_{\boldsymbol{\phi}}\mathcal{L}(\boldsymbol{\phi})\big|_{\boldsymbol{\phi}=\boldsymbol{\theta}}$, points in a direction that moves $\pi_{\boldsymbol{\phi}}$ toward $\pi_{\mathcal{D}}$, so a step in this direction decreases the probabilities of actions that are under-sampled in $\mathcal{D}$ w.r.t. $\pi_{\boldsymbol{\theta}}$ and increases those of over-sampled actions. Therefore, taking a step in the *opposite* direction increases the probabilities of under-sampled actions and decreases those of over-sampled actions, providing a way to reduce sampling error without ever computing $\pi_{\mathcal{D}}$. Thus, ROS performs a single step of descent on the *negative* log-likelihood

$$-\nabla_{\boldsymbol{\phi}}\mathcal{L}(\boldsymbol{\phi})\big|_{\boldsymbol{\phi}=\boldsymbol{\theta}} = \sum\nolimits_{(\boldsymbol{s},\boldsymbol{a}) \in \mathcal{D}} -\nabla_{\boldsymbol{\phi}} \log \pi_{\boldsymbol{\phi}}(\boldsymbol{a}|\boldsymbol{s})|_{\boldsymbol{\phi}=\boldsymbol{\theta}}, \tag{7}$$

at each timestep to increase the probability of under-sampled actions. In theory and in low-dimensional RL policy evaluation tasks such as CartPole-v1 (Brockman et al., 2016), this update was shown to improve the rate at which the empirical data distribution converges to the on-policy distribution—even when the empirical data distribution contains off-policy data. Unfortunately, two main challenges render ROS unsuitable for Line 10 in Algorithm 1.

**Challenge 1: Destructively large policy updates.** Since $\mathcal{D}$ may contain data collected by old target policies, some samples in $\mathcal{D}$ may be very off-policy w.r.t. the current target policy such that $\log \pi_{\boldsymbol{\phi}}(\boldsymbol{a}|\boldsymbol{s})$ is

---

[3]The advantage estimates in historic data correspond to historic policies and are thus biased advantage estimates of the current policy. In Section 7, we provide additional detail on how to mitigate this bias with generalized advantage estimation (GAE).

large and negative. Since $\nabla_{\boldsymbol{\phi}} \log \pi_{\boldsymbol{\phi}}(\boldsymbol{a}|\boldsymbol{s})$ increases in magnitude as $\pi_{\boldsymbol{\phi}}(\boldsymbol{a}|\boldsymbol{s})$ approaches zero, those off-policy samples can produce destructively large updates.




**Challenge 2: Improper handling of continuous actions.** In continuous-action tasks, ROS may produce behavior policies that *increase* sampling error. Continuous policies $\pi_{\boldsymbol{\theta}}(\boldsymbol{a}|\boldsymbol{s})$ are typically parameterized as Gaussians $\mathcal{N}(\boldsymbol{\mu}(\boldsymbol{s}), \Sigma(\boldsymbol{s}))$ with mean $\boldsymbol{\mu}(\boldsymbol{s})$ and diagonal covariance matrix $\Sigma(\boldsymbol{s})$. Since actions in the tail of the Gaussian far from the mean will generally be under-sampled, the ROS update will continually push the components of $\boldsymbol{\mu}(\boldsymbol{s})$ towards $\pm\infty$ and the diagonal components of $\Sigma(\boldsymbol{s})$ towards 0 to increase the probability of sampling these actions. The result is a degenerate behavior policy that is so far from the target policy that sampling from it increases sampling error. We illustrate this scenario with 1-dimensional continuous actions in Fig. 8 of Appendix C. In the next section, we discuss how to address both of these challenges.




---
**Algorithm 2** PROPS Update

---
1: **Inputs:** Target policy parameters $\boldsymbol{\theta}$, buffer $\mathcal{D}$, target KL $\delta_{\text{PROPS}}$, clipping coefficient $\epsilon_{\text{PROPS}}$, regularizer coefficient $\lambda$, n_epoch, n_minibatch.
2: **Output:** Behavior policy parameters $\boldsymbol{\phi}$.
3: $\boldsymbol{\phi} \leftarrow \boldsymbol{\theta}$
4: **for** epoch $i = 1, 2, \ldots,$ n_epoch **do**
5:     **for** minibatch $j = 1, 2, \ldots,$ n_minibatch **do**
6:         Sample minibatch $\mathcal{D}_j \sim \mathcal{D}$
7:         Update $\boldsymbol{\phi}$ with a step of gradient ascent on loss

$$\frac{1}{|\mathcal{D}_j|} \sum_{(\boldsymbol{s},\boldsymbol{a})\in\mathcal{D}_j} \mathcal{L}_{\text{PROPS}}(\boldsymbol{s}, \boldsymbol{a}, \boldsymbol{\phi}, \boldsymbol{\theta}, \epsilon_{\text{PROPS}}, \lambda)$$

8:         **if** $D_{\text{KL}}(\pi_{\boldsymbol{\theta}}||\pi_{\boldsymbol{\phi}}) > \delta_{\text{PROPS}}$ **then**
9:             **return** $\boldsymbol{\phi}$
10: **return** $\boldsymbol{\phi}$

---




## 5.2 Proximal Robust On-Policy Sampling

To address the challenges ROS faces in continuous control, we propose a new behavior policy update. To address Challenge 1, first observe that the gradient of the ROS loss $\nabla_{\boldsymbol{\phi}}\mathcal{L} = \nabla_{\boldsymbol{\phi}} \log \pi_{\boldsymbol{\phi}}(\boldsymbol{a}|\boldsymbol{s})|_{\boldsymbol{\phi}=\boldsymbol{\theta}}$ is equivalent to the policy gradient (Eq. 2) with $A^{\pi_{\boldsymbol{\theta}}}(\boldsymbol{s}, \boldsymbol{a}) = -1, \forall(\boldsymbol{s}, \boldsymbol{a})$. Since the clipped surrogate objective of PPO (Eq. 3) prevents destructively large updates in on-policy policy gradient learning, we can use a similar clipped objective in place of the ROS objective to prevent destructive behavior policy updates:

$$\mathcal{L}_{\text{CLIP}}(\boldsymbol{s}, \boldsymbol{a}, \boldsymbol{\phi}, \boldsymbol{\theta}, \epsilon_{\text{PROPS}}) = \min\left[ -\frac{\pi_{\boldsymbol{\phi}}(\boldsymbol{a}|\boldsymbol{s})}{\pi_{\boldsymbol{\theta}}(\boldsymbol{a}|\boldsymbol{s})}, -\texttt{clip}\left( \frac{\pi_{\boldsymbol{\phi}}(\boldsymbol{a}|\boldsymbol{s})}{\pi_{\boldsymbol{\theta}}(\boldsymbol{a}|\boldsymbol{s})}, 1 - \epsilon_{\text{PROPS}}, 1 + \epsilon_{\text{PROPS}} \right) \right]. \tag{8}$$

Table 2 in Appendix C summarizes the behavior of $\mathcal{L}_{\text{CLIP}}$. Intuitively, this objective is equivalent to the PPO objective (Eq. 3) with $A(\boldsymbol{s}, \boldsymbol{a}) = -1, \forall(\boldsymbol{s}, \boldsymbol{a})$ and incentivizes the agent to decrease the probability of observed actions by at most a factor of $1 - \epsilon_{\text{PROPS}}$. Let $g(\boldsymbol{s}, \boldsymbol{a}, \boldsymbol{\phi}, \boldsymbol{\theta}) = \frac{\pi_{\boldsymbol{\phi}}(\boldsymbol{a}|\boldsymbol{s})}{\pi_{\boldsymbol{\theta}}(\boldsymbol{a}|\boldsymbol{s})}$. When $g(\boldsymbol{s}, \boldsymbol{a}, \boldsymbol{\phi}, \boldsymbol{\theta}) < 1 - \epsilon_{\text{PROPS}}$, this objective is clipped at $-(1 - \epsilon_{\text{PROPS}})$. The loss gradient $\nabla_{\boldsymbol{\phi}}\mathcal{L}_{\text{CLIP}}$ becomes zero, and the $(\boldsymbol{s}, \boldsymbol{a})$ pair has no effect on the policy update. When $g(\boldsymbol{s}, \boldsymbol{a}, \boldsymbol{\phi}, \boldsymbol{\theta}) > 1 - \epsilon_{\text{PROPS}}$, clipping does not apply, and the gradient $\nabla_{\boldsymbol{\phi}}\mathcal{L}_{\text{CLIP}}$ points in a direction that decreases the probability of $\pi_{\boldsymbol{\phi}}(\boldsymbol{a}|\boldsymbol{s})$. As in the PPO update, this clipping mechanism avoids destructively large policy updates and permits us to perform multiple epochs of minibatch updates with the same batch of data.

To address the second challenge and prevent degenerate behavior policies, we introduce an auxiliary loss that incentivizes the agent to minimize the KL divergence between the behavior policy and target policy at states in the observed data. The full PROPS objective is then:

$$\mathcal{L}_{\text{PROPS}}(\boldsymbol{s}, \boldsymbol{a}, \boldsymbol{\phi}, \boldsymbol{\theta}, \epsilon_{\text{PROPS}}, \lambda) = \mathcal{L}_{\text{CLIP}}(\boldsymbol{s}, \boldsymbol{a}, \boldsymbol{\phi}, \boldsymbol{\theta}) - \lambda D_{\text{KL}}(\pi_{\boldsymbol{\theta}}(\cdot|\boldsymbol{s})||\pi_{\boldsymbol{\phi}}(\cdot|\boldsymbol{s})) \tag{9}$$

where $\lambda$ is a regularization coefficient quantifying a trade-off between maximizing $\mathcal{L}_{\text{PROPS}}$ and minimizing $D_{\text{KL}}$. Algorithm 2 provides pseudocode for the PROPS update. Like ROS, we set the behavior policy parameters $\boldsymbol{\phi}$ equal to the target policy parameters at the start of each behavior update, and then make a local adjustment to $\boldsymbol{\phi}$ to increase the probabilities of under-sampled actions. We stop the PROPS update early when $D_{\text{KL}}(\pi_{\boldsymbol{\theta}}||\pi_{\boldsymbol{\phi}})$ reaches a chosen threshold $\delta_{\text{PROPS}}$. This technique further safeguards against large policy updates and is used in widely adopted implementations of PPO (Raffin et al., 2021; Liang et al.,

2018).[4] In Appendix B, we provide theoretical intuition for the relationship between different PROPS hyperparameters. PROPS enables us to efficiently learn a behavior policy that keeps the distribution of data in the buffer close to the expected distribution of the target policy.

## 6 Experiments

The central goal of our work is to understand if reducing sampling error with adaptive, off-policy sampling results in more data-efficient on-policy policy gradient learning. Towards this goal, we design experiments on continuous-state and continuous-action MuJoCo benchmark tasks (Brockman et al., 2016) and a tabular 5x5 GridWorld task (Fig. 2a) to answer the following questions:

**Q1:** Does PROPS achieve lower sampling error than on-policy sampling during RL training?

**Q2:** Does PROPS improve the data efficiency of on-policy policy gradient algorithms? That is, does PROPS reduce the number of environment interactions required to achieve returns comparable to on-policy sampling?

### 6.1 Sampling Error Metrics

In GridWorld, we compute sampling error as the total variation (TV) distance between the empirical state-action visitation $d_{\mathcal{D}}(\boldsymbol{s}, \boldsymbol{a})$ distribution—denoting the proportion of times $(\boldsymbol{s}, \boldsymbol{a})$ appears in buffer $\mathcal{D}$—and the true state-action visitation distribution under the agent's policy: $\sum_{(\boldsymbol{s},\boldsymbol{a})\in\mathcal{D}} |d_{\pi_{\boldsymbol{\theta}}}(\boldsymbol{s}, \boldsymbol{a}) - d_{\mathcal{D}}(\boldsymbol{s}, \boldsymbol{a})|$. Since it is straightforward to compute the true policy gradient in the GridWorld task, we additionally investigate how sampling error reduction affects gradient estimation by measuring the cosine similarity between the empirical policy gradient $\nabla_{\boldsymbol{\theta}} \widehat{J}(\boldsymbol{\theta})$ and the true policy gradient. As the empirical gradient aligns more closely with the true gradient, the cosine similarity approaches 1. In continuous MuJoCo tasks where it is difficult to compute $d_{\mathcal{D}}(\boldsymbol{s}, \boldsymbol{a})$, we compute sampling error as the KL-divergence $D_{\mathrm{KL}}(\pi_{\mathcal{D}}||\pi_{\boldsymbol{\theta}})$ between the empirical policy $\pi_{\mathcal{D}}$ and the target policy $\pi_{\boldsymbol{\theta}}$, which is the primary metric Zhong et al. (2022) used to show ROS reduces sampling error:

$$D_{\mathrm{KL}}(\pi_{\mathcal{D}}||\pi_{\boldsymbol{\theta}}) = \mathbb{E}_{\boldsymbol{s}\sim\mathcal{D},\boldsymbol{a}\sim\pi_{\mathcal{D}}(\cdot|\boldsymbol{s})}\left[\log\left(\frac{\pi_{\mathcal{D}}(\boldsymbol{a}|\boldsymbol{s})}{\pi_{\boldsymbol{\theta}}(\boldsymbol{a}|\boldsymbol{s})}\right)\right]. \tag{10}$$

We estimate $\pi_{\mathcal{D}}$ as the maximum likelihood estimate under data in the buffer via stochastic gradient ascent. More concretely, we let $\boldsymbol{\theta}'$ be the parameters of a neural network with the same architecture as $\pi_{\boldsymbol{\theta}}$ and then compute:

$$\boldsymbol{\theta}_{\mathrm{MLE}} = \arg\max_{\boldsymbol{\theta}'} \sum_{(\boldsymbol{s},\boldsymbol{a})\in\mathcal{D}} \log \pi_{\boldsymbol{\theta}'}(\boldsymbol{a}|\boldsymbol{s}) \tag{11}$$

using stochastic gradient ascent. After computing $\boldsymbol{\theta}_{\mathrm{MLE}}$, we then estimate sampling error using the Monte Carlo estimator:

$$D_{\mathrm{KL}}(\pi_{\mathcal{D}}||\pi_{\boldsymbol{\theta}}) \approx \sum_{(\boldsymbol{s},\boldsymbol{a})\in\mathcal{D}} \left(\log \pi_{\boldsymbol{\theta}_{\mathrm{MLE}}}(\boldsymbol{a}|\boldsymbol{s}) - \log \pi_{\boldsymbol{\theta}}(\boldsymbol{a}|\boldsymbol{s})\right). \tag{12}$$

### 6.2 Correcting Sampling Error for a Fixed Target Policy

We first study how quickly PROPS decreases sampling error when the target policy is fixed. This setting is similar to the policy evaluation setting considered by Zhong et al. (2022). As such, we provide two core baselines for comparison: on-policy sampling and ROS. In GridWorld, we additionally include an oracle sampling method that always selects the most under-sampled state–action pair $(\boldsymbol{s}^*, \boldsymbol{a}^*) = \arg\max_{(\boldsymbol{s},\boldsymbol{a})} d_{\pi_{\boldsymbol{\theta}}}(\boldsymbol{s}, \boldsymbol{a}) - d_{\mathcal{D}}(\boldsymbol{s}, \boldsymbol{a})$. This sampler is not realizable in standard RL settings because it assumes the ability to reset the agent to an arbitrary state, but it provides a useful reference point for the minimum sampling error achievable for a given number of samples. We provide details on hyperparameter tuning in Appendix D.2.

---

[4]The KL regularizer Eq. 9 and the KL cutoff rule $D_{KL}(\pi_{\theta}||\pi_{\phi}) > \delta_{\mathrm{PROPS}}$ serve different purposes. KL regularization limits how far the behavior policy can move from the target policy *after each gradient step*, while the cutoff rule prevents the behavior policy from drifting too far *across multiple epochs of minibatch updates*, even if each step individually is small

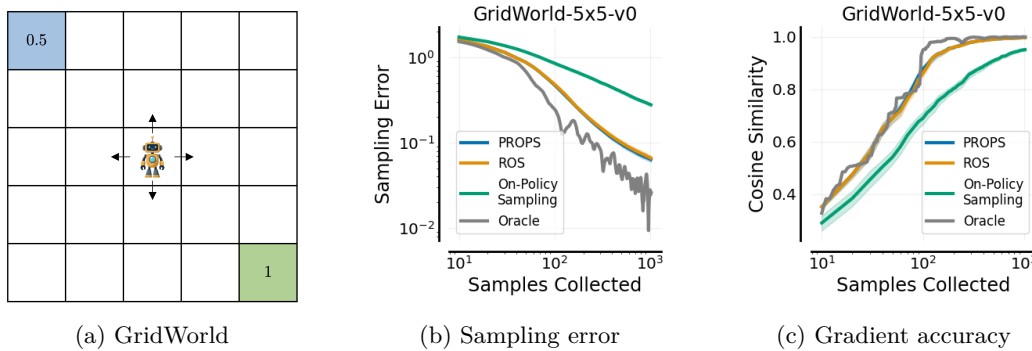

(a) GridWorld        (b) Sampling error        (c) Gradient accuracy

Figure 2: **(a)** A GridWorld task in which the agent receives reward $+1$ upon reaching the bottom right corner (the optimal goal), a reward of $+0.5$ upon reaching the top left corner (the suboptimal goal), and a reward of $-0.01$. The agent always starts in the center of the grid. Under an initially uniform policy, the agent visits both goals with equal probability, and thus the true policy gradient increases the probability of reaching the optimal goal. However, sampling error can yield an empirical gradient that increases the probability of reaching the suboptimal goal and cause the agent to converge suboptimally. To converge optimally, the agent must have low sampling error. **(b, c)** PROPS reduces sampling error and achieves more accurate gradients faster than on-policy sampling. Solid curves denote means over 50 seeds. Shaded regions denote 95% bootstrap confidence belts.

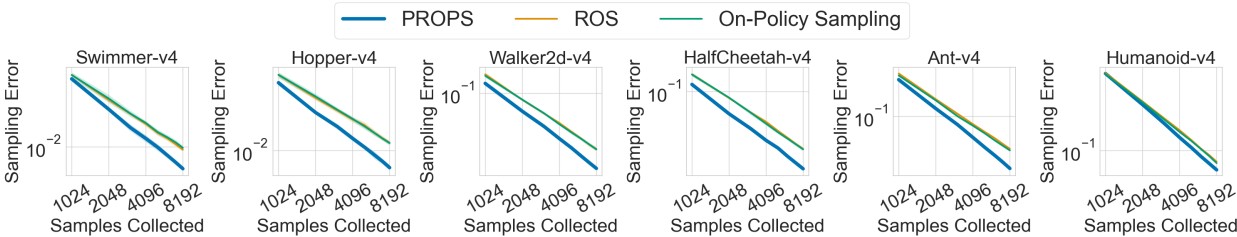

Figure 3: Sampling error with a fixed, randomly initialized target policy. Solid curves denote the mean over 5 seeds. Shaded regions denote 95% confidence belts.

**Results.** We collect 8192 samples with a randomly initialized policy and compute sampling error throughout data collection. As shown in Fig. 2b and 2c, in GridWorld, PROPS decreases sampling error faster than on-policy sampling, resulting in more accurate policy gradient estimates. PROPS and ROS perform similarly, though this behavior is expected: in a tabular setting with a fixed target policy, there is no off-policy data in the buffer, so we do not encounter Challenge 1 and 2 described in the previous section. In Appendix D.1, we empirically demonstrate that PROPS is unbiased and has lower variance than on-policy sampling. The oracle sampler attains lower sampling error and higher gradient accuracy than PROPS, as expected, since it assumes access to state resets.[5]

In continuous MuJoCo tasks where Challenge 2 arises, PROPS decreases sampling error faster than on-policy sampling and ROS (Fig. 3). In fact, ROS shows no improvement over on-policy sampling in all MuJoCo tasks. This limitation of ROS is unsurprising, as Zhong et al. (2022) showed that ROS struggled to reduce sampling error even in low-dimensional continuous-action tasks. Since batch sizes in the range of 2048–8192 are standard for RL in these tasks, ROS alone will not be able to improve data efficiency during RL. In

---

[5] Sampling error does not decrease monotonically under the oracle sampler. With $T$ samples, the expected number of visits to each $(s, a)$ under on-policy sampling is $T \cdot d_{\pi_\theta}(s, a)$, but these quantities are generally not integers, so the oracle sampler must therefore choose the nearest integer allocation, which forces some pairs to be over- or under-sampled. Depending on $T$, this rounding may align more or less closely with the target distribution, leading to fluctuations in the minimum achievable TV distance. For a concrete illustration, suppose there are $n$ state–action pairs and $d_{\pi_\theta}$ is uniform. When $T$ is a multiple of $n$, the oracle can allocate counts evenly across all pairs; for other values (e.g., $n + 1, \ldots, 2n - 1$), at least one pair must receive an extra or missing count, necessarily increasing sampling error.

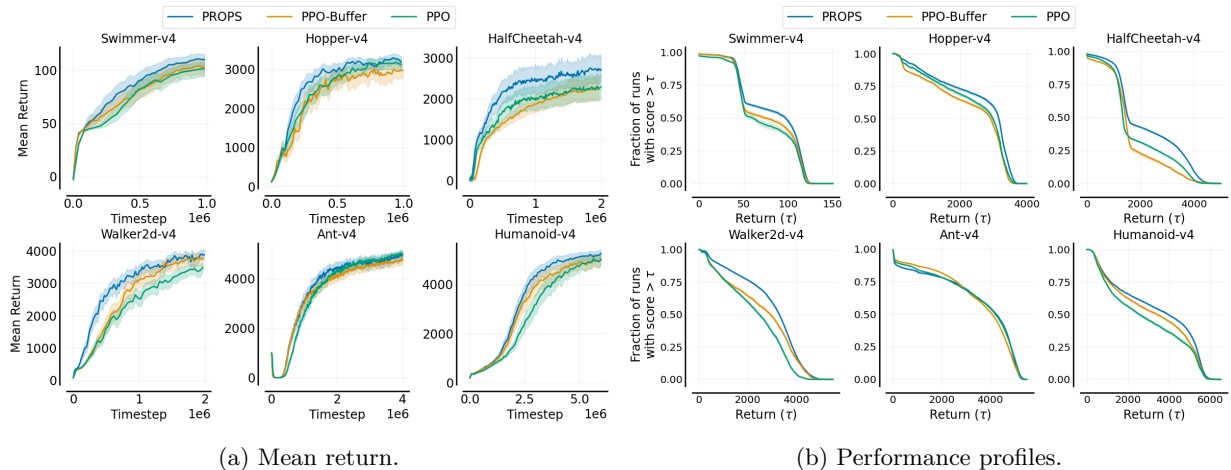

(a) Mean return.

(b) Performance profiles.

Figure 4: **(a)** Mean returns over 50 seeds. **(b)** Performance profiles over 50 seeds. Higher values correspond to more reliable convergence to high-return policies. Shaded regions denote 95% bootstrap confidence intervals.

| Environment | 33% of training budget | | | 66% of training budget | | | 100% of training budget | | |
|---|---|---|---|---|---|---|---|---|---|
| | PROPS | PPO-Buffer | PPO | PROPS | PPO-Buffer | PPO | PROPS | PPO-Buffer | PPO |
| Swimmer-v4 | **70.5** | 64.8 | 55.6 | **97.4** | 90.5 | 90.9 | **109.7** | 102.9 | 100.9 |
| Hopper-v4 | **2675** | 2307 | 2347 | **3144** | 2801 | 2946 | **3253** | 2967 | 3148 |
| HalfCheetah-v4 | **2292** | 1546 | 1810 | **2591** | 2032 | 2133 | **2751** | 2237 | 2270 |
| Walker2d-v4 | **2880** | 2107 | 1989 | **3590** | 3336 | 2858 | **3908** | 3737 | 3397 |
| Ant-v4 | **3519** | 3493 | 3484 | **4668** | 4419 | 4589 | 4930 | 4741 | **5014** |
| Humanoid-v4 | **2270** | 1938 | 1376 | **4891** | 4449 | 4067 | **5184** | 4958 | 4955 |

Table 1: Return achieved at different points throughout training. We use color to indicate statistical significance at a 95% confidence level according to a paired t-test. Green indicates PROPS outperforms PPO and PPO-Buffer with significance. Grey indicates PROPS outperforms only PPO with significance. Blue indicates PROPS outperforms only PPO-Buffer with significance. **Bold** values indicate the largest average return. PROPS achieves a larger return than PPO and PPO-Buffer at all points except at the end of training in Ant-v4, where PPO achieves a slightly larger (but similar) return.

Appendix D.2, we include similar experiments with expert target policies as well as ablations on PROPS's objective clipping and regularization. Results with expert target policies are qualitatively similar to Fig. 3, and we observe that clipping and regularization both individually help reduce sampling error.

## 6.3 Correcting Sampling Error During RL Training

We are ultimately interested in understanding how replacing on-policy sampling with PROPS affects the data efficiency of on-policy learning. In the following experiments, we train RL agents with PROPS and on-policy sampling to evaluate (1) the data efficiency of training, (2) the distribution of returns achieved at the end of training, and (3) the sampling error throughout training. We use the same sampling error metrics described in the previous section and measure data efficiency as the return achieved within a fixed training budget. Since ROS is computationally expensive and fails to reduce sampling error in MuJoCo tasks even with a fixed policy, we omit it from MuJoCo experiments.

**Experimental setup.** We use PPO (Schulman et al., 2017) to update the target policy and consider two baseline methods for providing data to compute PPO updates: (1) vanilla PPO with on-policy sampling, and (2) PPO with on-policy sampling using a buffer of size $b$ (PPO-Buffer). PPO-Buffer is a naive method for improving data efficiency of on-policy algorithms by reusing off-policy data collected by old target policies as if it were on-policy data. Although PPO-Buffer computes biased gradients, it has been successfully applied

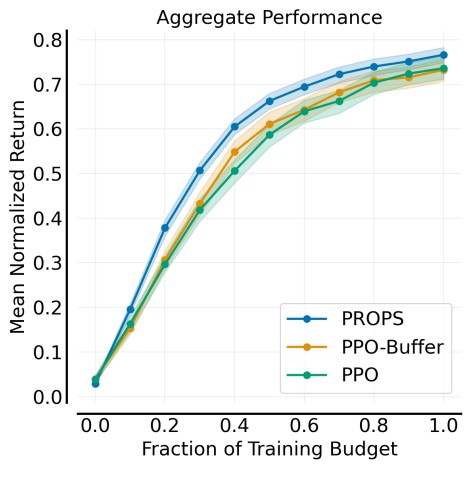
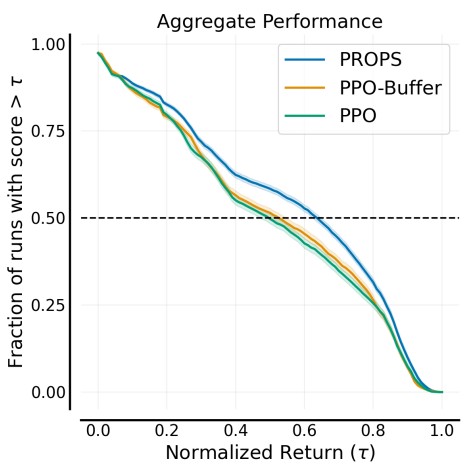

(a) Mean normalized return (aggregated).   (b) Performance profiles (aggregated).

Figure 5: Mean normalized return and performance profiles aggregated over all six MuJoCo tasks. We compute normalized returns as $\frac{R_{\max}-R_t}{R_{\max}}$, where $R_{\max}$ is the maximum return achieved by any algorithm in a particular task, and $R_t$ is the return at timestep $t$. Solid curves denote the mean over 50 seeds per task (300 seeds total). Shaded regions denote 95% bootstrap confidence belts.

in difficult learning tasks (Berner et al., 2019). Since PROPS and PPO-Buffer have access to the same amount of data for each policy update, any performance difference between these two methods arises from differences in how they sample actions during data collection.

In MuJoCo experiments, we set $b = 2$ such that agents retain each batch of data for one extra iteration before discarding it. In GridWorld, we use $b = 1$ and discard all historic data. Since PROPS and PPO-Buffer compute target policy updates with $b$ times as much learning data as PPO, we integrate this extra data by increasing the minibatch size for target and behavior policy updates by a factor of $b$. Further experimental details including hyperparameter tuning are described in Appendix F. For MuJoCo tasks, we plot the mean return throughout training as well as the distribution of returns achieved at the end of training (*i.e.*, the performance profile) (Agarwal et al., 2021). For GridWorld, we plot the fraction of times agents find the optimal goal (*i.e.*, the success rate).

**Results.** Fig. 6a shows that on-policy sampling has approximately a 77% success rate on GridWorld, whereas PROPS and ROS achieve a 100% success rate. Fig. 4a and Table 1 show that PROPS can achieve the same returns as PPO and PPO-Buffer with fewer environment interactions. For instance, in Humanoid-v4 and Hopper-v4, PROPS closely matches the final return of PPO with 33% fewer interactions. To provide a more compact and readable summary of performance throughout training, we also report results aggregated across all six MuJoCo tasks in Fig. 5a. PROPS matches the normalized return of PPO and PPO-Buffer using only 60% of the training budget. Moreover, the performance

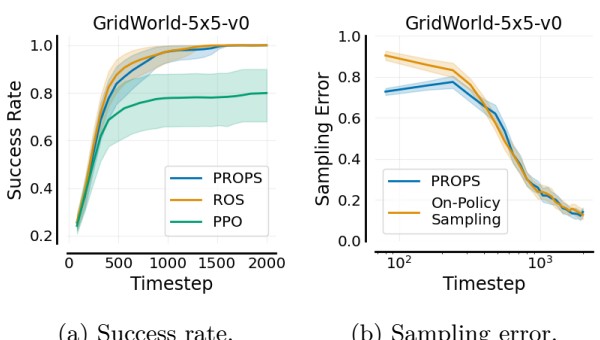

(a) Success rate.   (b) Sampling error.

Figure 6: GridWorld RL experiments over 50 seeds.

profiles of PROPS in Fig. 4b and Fig. 5b are almost always above the performance profiles of PPO and PPO-Buffer, indicating that any given run of PROPS is more likely to obtain a higher return than PPO-Buffer. Thus, we affirmatively answer **Q2** posed at the start of this section: PROPS increases the fraction of training runs with high return and increases data efficiency.

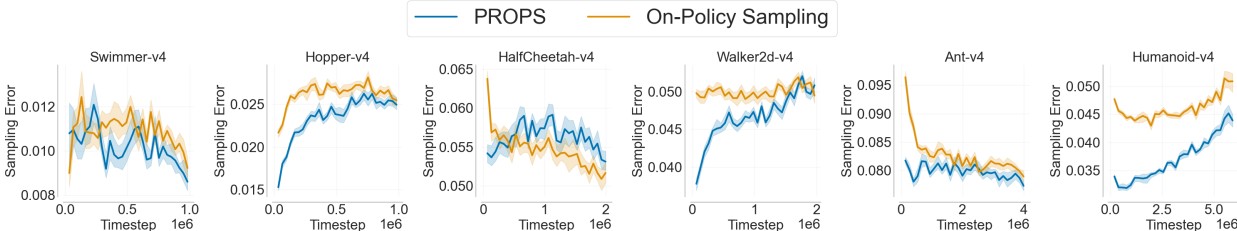

Figure 7: Sampling error throughout RL training. Solid curves denote the mean over 5 seeds. Shaded regions denote 95% confidence belts.

Having established that PROPS improves data efficiency, we now investigate if PROPS is appropriately adjusting the data distribution of the buffer by comparing the sampling error achieved throughout training with PROPS and PPO-Buffer. Training with PROPS produces a different sequence of target policies than training with PPO-Buffer produces. To provide a fair comparison, we compute sampling error for PPO-Buffer using the target policy sequence produced by PROPS. More concretely, we fill a second buffer with on-policy samples collected by the *target policies* produced while training with PROPS and then compute the sampling error using data in this buffer.

As shown in Fig. 7, PROPS achieves lower sampling error than on-policy sampling throughout RL training in 5 out of 6 tasks. In HalfCheetah-v4, PROPS decreases sampling error only in the first 400k timesteps, but nevertheless improves data efficiency. This result likely reflects our hyperparameter tuning procedure in which we selected hyperparameters yielding the largest return (Appendix F). Although lower sampling error intuitively correlates with increased data efficiency, it is nevertheless possible to achieve high return without reducing sampling error. Alternatively, the early reduction in sampling error itself may be sufficient to explain the observed efficiency gains. In GridWorld, PROPS and ROS reduce sampling error in the first 300 steps and closely match on-policy sampling afterwards. We use a batch size of 80 in these experiments, and as the target policy becomes more deterministic, larger batch sizes are needed to observe differences between PROPS and on-policy sampling.[6]

We ablate the effects of the clipping coefficient $\epsilon_{\mathrm{PROPS}}$, regularization coefficient $\lambda$, and buffer size $b$ in Appendix D.3. Without clipping or without regularization, PROPS often achieves greater sampling error than on-policy sampling, indicating that both help to keep sampling error low. Moreover, data efficiency generally decreases when we remove clipping or regularization, showing both are essential to PROPS. We find that data efficiency may decrease with a larger buffer size. Intuitively, the more historic data kept around, the more data that must be collected to impact the aggregate data distribution. Last, we show that PROPS is robust to hyperparameter choices using a sensitivity analysis in Appendix G. Thus, we affirmatively answer **Q1** posed at the start of this section: PROPS achieves lower sampling error than on-policy sampling when the target policy is fixed and during RL training.

## 7 Discussion

This work has shown that adaptive, off-policy sampling can reduce sampling error in data collected throughout RL training and increase the data efficiency of on-policy policy gradient algorithms. In this section, we discuss limitations of our work and present opportunities for future research.

**Convergence of PROPS.** PROPS builds upon the ROS algorithm of Zhong et al. (2022). While Zhong et al. (2022) focused on theoretical analysis and policy evaluation in low-dimensional domains, we chose to focus on empirical analysis with policy learning in standard RL benchmarks. Proposition 1 and previous convergence results by Zhong et al. (2022) focus on tabular MDPs and provide conceptual grounding for PROPS, but this theory does not immediately extend to continuous MDPs. An important direction for future work would be theoretical analysis of PROPS, in particular whether PROPS also enjoys the same faster convergence rate that was shown for ROS relative to on-policy sampling in continuous MDPs.

---

[6]As a policy becomes deterministic, sampling error approaches zero, so there is less sampling error for PROPS to correct.

**Advantage Estimates of Historic Data.** While PROPS can reduce sampling error in off-policy historic data, the advantage estimates associated with this data still correspond to historic policies and are thus biased advantage estimates of the current policy. PPO computes advantages using GAE (Schulman et al., 2016) with parameter $\lambda \in [0, 1]$ controlling how much the advantage estimates weight multi-step Monte Carlo returns versus value function predictions. In practice, PPO uses a relatively large $\lambda$, typically $\lambda = 0.95$, which heavily weights Monte Carlo returns. Since these returns are from trajectories generated by a behavior policy correcting sampling error w.r.t. a historic policy, the returns correspond to the historic policy, not the current one. In other words, PROPS corrects the off-policy bias of the $(\boldsymbol{s}, \boldsymbol{a})$ distribution but not that of the advantage estimates. Empirically, this bias appears minor, as PROPS still improves data efficiency. Nevertheless, it can be mitigated by decreasing $\lambda$ so GAE places greater weight on value function predictions, which are updated to reflect the current policy and will thus produce advantage estimates that better align with the current policy.

**Advantage-Weighted Sampling Error Correction.** A limitation of PROPS is that the update indiscriminately increases the probability of under-sampled actions without considering their importance in gradient computation. For instance, if an under-sampled action has zero advantage, it has no impact on the gradient and need not be sampled. An interesting direction for future work could be to prioritize correcting sampling error for $(\boldsymbol{s}, \boldsymbol{a})$ that have the largest influence on the gradient estimate, *i.e.*, those with large advantages (positive or negative).

**Sampling Error Correction for Off-Policy RL.** Beyond these more immediate directions, our work opens up other opportunities for future research. A less obvious feature of the PROPS behavior policy update is that it can be used to track the empirical data distribution of *any* desired policy, not only that of the current policy. This feature means PROPS has the potential to be integrated into off-policy RL algorithms and used so that the empirical distribution more closely matches a desired exploration distribution. Thus, PROPS could be used to perform focused exploration without explicitly tracking state and action counts.

## 8 Conclusion

In this work, we ask if adaptive, *off-policy* action sampling can produce data that more closely matches the expected on-policy data distribution and improve the data efficiency of on-policy policy gradient methods. To answer this question, we introduce Proximal Robust On-policy Sampling (PROPS), a method that periodically updates a data collection behavior policy to increase the probability of sampling actions that are currently under-sampled with respect to the on-policy distribution. Furthermore, rather than discarding collected data after every policy update, PROPS permits more data-efficient on-policy learning by using data collection to adjust the distribution of previously collected data to be approximately on-policy. We replace on-policy sampling with PROPS to collect data for the popular PPO algorithm and empirically demonstrate that PROPS produces data that more closely matches the expected on-policy distribution and yields more data-efficient learning compared to on-policy sampling.

## Acknowledgments

We thank Adam Labiosa, Abhinav Harish, and Brahma Pavse for discussion and comments that improved the final version of this work. This work took place in the Prediction and Action Lab (PAL) at the University of Wisconsin – Madison. PAL research is supported by the National Science Foundation (IIS-2410981) and the Wisconsin Alumni Research Foundation.

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

## Appendix Contents

## A  Core Theoretical Results

In this section, we present the proof of Proposition 1. We use $d_m$, $\pi_m$, and $p_m$ as the empirical state visitation distribution, empirical policy, and empirical transition probabilities after $m$ state-action pairs have been taken, respectively. That is, $d_m(s)$ is the proportion of the $m$ states that are $s$, $\pi_m(a|s)$ is the proportion of the time that action $a$ was observed in state $s$, and $p_m(s'|s,a)$ is the proportion of the time that the state changed to $s'$ after action $a$ was taken in state $s$.

**Proposition 2.** *Assume that data is collected with an adaptive behavior policy that always takes the most under-sampled action in each state, $s$, with respect to policy $\pi$, i.e., $a \leftarrow \arg\max_{a'}(\pi(a'|s) - \pi_m(a'|s))$. We further assume that $\mathcal{S}$ and $\mathcal{A}$ are finite and that the Markov chain induced by $\pi$ is irreducible. Then we have that the empirical state visitation distribution, $d_m$, converges to the state distribution of $\pi$, $d_\pi$, with probability 1:*

$$\forall s, \lim_{m \to \infty} d_m(s) = d_\pi(s).$$

*Proof.* The proof of this theorem builds upon Lemma 1 and 2 by Zhong et al. (2022). Note that these lemmas superficially concern the ROS method whereas we are interested in data collection by taking the most under-sampled action at each step. However, as stated in the proof by Zhong et al. (2022), these methods are equivalent under an assumption that they make about the step-size parameter of the ROS method. Thus, we can immediately adopt these lemmas for this proof.

Under Lemma 1 of Zhong et al. (2022), we have that $\lim_{m\to\infty} \pi_m(a|s) = \pi(a|s)$ for any state $s$ under this adaptive data collection procedure. We then have the following $\forall s$:

$$\lim_{m\to\infty} d_m(s) \overset{(a)}{=} \lim_{m\to\infty} \sum_{\tilde{s}} \sum_{\tilde{a}} p_m(s|\tilde{s},\tilde{a})\pi_m(\tilde{a}|\tilde{s})d_m(\tilde{s})$$

$$= \sum_{\tilde{s}} \sum_{\tilde{a}} \lim_{m\to\infty} p_m(s|\tilde{s},\tilde{a})\pi_m(\tilde{a}|\tilde{s})d_m(\tilde{s})$$

$$= \sum_{\tilde{s}} \sum_{\tilde{a}} \lim_{m\to\infty} p_m(s|\tilde{s},\tilde{a}) \lim_{m\to\infty} \pi_m(\tilde{a}|\tilde{s}) \lim_{m\to\infty} d_m(\tilde{s})$$

$$\overset{(b)}{=} \sum_{\tilde{s}} \sum_{\tilde{a}} p(s|\tilde{s},\tilde{a})\pi(\tilde{a}|\tilde{s}) \lim_{m\to\infty} d_m(\tilde{s}).$$

Here, (a) follows from the fact that the empirical frequency of state $s$ can be obtained by considering all possible transitions that lead to $s$. The last line, (b), holds with probability 1 by the strong law of large numbers and Lemma 2 of Zhong et al. (2022).

We now have a system of $|\mathcal{S}|$ variables and $|\mathcal{S}|$ linear equations. Define variables $x(s) \coloneqq \lim_{m \to \infty} d_m(s)$ and let $\boldsymbol{x} \in \mathbf{R}^{|\mathcal{S}|}$ be the vector of these variables. We then have $x = P^\pi x$ where $P^\pi \in \mathbf{R}^{|\mathcal{S}| \times |\mathcal{S}|}$ is the transition matrix of the Markov chain induced by running policy $\pi$. Assuming that this Markov chain is irreducible, $d_\pi$ is the unique solution to this system of equations and hence $\lim_{m \to \infty} d_m(s) = d_\pi(s), \forall s$.

$\square$

## B   Additional Theoretical Results

In this section, we provide additional theory to describe the relationship between different hyperparameters in PROPS:

1. The amount of sampling error in previously collected data and the size of behavior policy updates.

2. The amount of historic data retained by an agent and the amount of additional data the behavior policy must collect to reduce sampling error.

For simplicity, we first focus on a simple bandit setting and then extend to a tabular RL setting.

Suppose we have already collected $m$ state-action pairs and these have been observed with empirical distribution $\pi_m(\boldsymbol{a})$. From what distribution should we sample an additional $k$ state-action pairs so that the empirical distribution over the $m + k$ samples is equal in expectation to $\pi_{\boldsymbol{\theta}}$?

**Proposition 3.** *Assume that $m$ actions have been collected by running some policy $\pi_{\boldsymbol{\theta}}(\boldsymbol{a})$ and $\pi_m(\boldsymbol{a})$ is the empirical distribution on this dataset. If we collect an additional $k$ state-action pairs using the following distribution, and if $(m + k)\pi_{\boldsymbol{\theta}}(\boldsymbol{a}) \geq m \cdot \pi_m(\boldsymbol{a})$, then the aggregate empirical distribution over the $m + k$ pairs is equal to $\pi_{\boldsymbol{\theta}}(\boldsymbol{a})$ in expectation:*

$$\pi_b(\boldsymbol{a}) \coloneqq \frac{1}{Z} \left[ \pi_{\boldsymbol{\theta}}(\boldsymbol{a}) + \frac{m}{k} \left( \pi_{\boldsymbol{\theta}}(\boldsymbol{a}) - \pi_m(\boldsymbol{a}) \right) \right]$$

*where $Z = \sum_{\boldsymbol{a} \in \mathcal{A}} \left[ \pi_{\boldsymbol{\theta}}(\boldsymbol{a}) + \frac{m}{k} \left( \pi_{\boldsymbol{\theta}}(\boldsymbol{a}) - \pi_m(\boldsymbol{a}) \right) \right]$ is a normalization coefficient.*

*Proof.* Observe that $(m + k)\pi_{\boldsymbol{\theta}}(\boldsymbol{a})$ is the expected number of times $\boldsymbol{a}$ is sampled under $\pi_{\boldsymbol{\theta}}$ after $m + k$ steps, $m \cdot \pi_m(\boldsymbol{a})$ is the number of times each $\boldsymbol{a}$ was sampled thus far, and $k \cdot \pi_b(\boldsymbol{a})$ is the expected number of times $\boldsymbol{a}$ is sampled under our behavior policy after $k$ steps. We want to choose $\pi_b(\boldsymbol{a})$ such that $(m + k)\pi_{\boldsymbol{\theta}}(\boldsymbol{a}) = m \cdot \pi_m(\boldsymbol{a}) + k \cdot \pi_b(\boldsymbol{a})$ in expectation.

$$(m + k)\pi_{\boldsymbol{\theta}}(\boldsymbol{a}) = k \cdot \pi_b(\boldsymbol{a}) + m \cdot \pi_m(\boldsymbol{a})$$
$$-k \cdot \pi_b(\boldsymbol{a}) = m \cdot \pi_m(\boldsymbol{a}) - (m + k)\pi_{\boldsymbol{\theta}}(\boldsymbol{a})$$
$$\pi_b(\boldsymbol{a}) = -\frac{m}{k} \pi_m(\boldsymbol{a}) + \left( \frac{m}{k} + 1 \right) \pi_{\boldsymbol{\theta}}(\boldsymbol{a})$$
$$= \pi_{\boldsymbol{\theta}}(\boldsymbol{a}) + \frac{m}{k} \left( \pi_{\boldsymbol{\theta}}(\boldsymbol{a}) - \pi_m(\boldsymbol{a}) \right)$$

Note that $\pi_b(\boldsymbol{a})$ will be a valid probability distribution after normalizing only if

$$\pi_{\boldsymbol{\theta}}(\boldsymbol{a}) + \frac{m}{k} \left( \pi_{\boldsymbol{\theta}}(\boldsymbol{a}) - \pi_m(\boldsymbol{a}) \right) \geq 0$$
$$\left( \frac{m}{k} + 1 \right) \pi_{\boldsymbol{\theta}}(\boldsymbol{a}) \geq \frac{m}{k} \pi_m(\boldsymbol{a})$$
$$(m + k) \pi_{\boldsymbol{\theta}}(\boldsymbol{a}) \geq m \cdot \pi_m(\boldsymbol{a}).$$

If $(m + k) \pi_{\boldsymbol{\theta}}(\boldsymbol{a}) < m \cdot \pi_m(\boldsymbol{a})$, then prior to collecting additional data with our behavior policy, $\boldsymbol{a}$ already appears in our data more times than it would in expectation after $m + k$ steps under $\pi_{\boldsymbol{\theta}}$. In other words,

we would need to collect more than $k$ additional samples to achieve zero sampling error (or *discard* some previously collected samples).

$\square$

**When sampling error is large, behavior policy updates must also be large.** Intuitively, the difference $\pi_{\boldsymbol{\theta}}(\boldsymbol{a}) - \pi_m(\boldsymbol{a})$ is the mismatch between the true and empirical visitation distributions, so adding this term to $d_{\pi_{\boldsymbol{\theta}}}$ adjusts $d_{\pi_{\boldsymbol{\theta}}}$ to reduce this mismatch. If $\pi_{\boldsymbol{\theta}}(\boldsymbol{a}) - \pi_m(\boldsymbol{a}) < 0$, then $\boldsymbol{a}$ is over-sampled w.r.t. $\pi_{\boldsymbol{\theta}}$, and $\pi_b$ will decrease the probability of sampling $\boldsymbol{a}$. If $\pi_{\boldsymbol{\theta}}(\boldsymbol{a}) - \pi_m(\boldsymbol{a}) > 0$, then $\boldsymbol{a}$ is under-sampled w.r.t. $\pi_{\boldsymbol{\theta}}$, and $\pi_b$ will increase the probability of sampling $\boldsymbol{a}$. When $|\pi_{\boldsymbol{\theta}}(\boldsymbol{a}) - \pi_m(\boldsymbol{a})|$ is small, the optimal $\pi_b(\boldsymbol{a})$ requires only a small adjustment from $\pi_{\boldsymbol{\theta}}$ (*i.e.*, a small update to the behavior policy is sufficient to reduce sampling error). When $|\pi_{\boldsymbol{\theta}}(\boldsymbol{a}) - \pi_m(\boldsymbol{a})|$ is large, the optimal $\pi_b(\boldsymbol{a})$ requires a large adjustment from $\pi_{\boldsymbol{\theta}}$ (*i.e.*, a large update to the behavior policy is needed to reduce sampling error). We can increase (or decrease) the target KL cutoff $\delta_{\mathrm{PROPS}}$ to allow for larger (or smaller) behavior updates.

**When we retain large amounts of historic data, the behavior policy must collect a large amount of additional data to reduce sampling error in the aggregate distribution.** The $\frac{m}{k}$ factor implies that how much we adjust $d_{\pi_{\boldsymbol{\theta}}}$ depends on how much data we have already collected ($m$) and how much additional data we will collect ($k$). If the $k$ additional samples to collect represent a small fraction of the aggregate $m + k$ samples (*i.e.* $k << m$), then $\frac{m}{k}$ is large, and the adjustment to $d_{\pi_{\boldsymbol{\theta}}}$ is large. This case generally arises when we retain more and more historic data. If the $k$ additional samples to collect represent a large fraction of the aggregate $m + k$ samples (*i.e.* $k >> m$), then $\frac{m}{k}$ is small, and the adjustment to $d_{\pi_{\boldsymbol{\theta}}}$ is small. This case generally arises when we retain little to no historic data.

The next proposition extends this analysis to the tabular RL setting.

**Proposition 4.** *Assume that $m$ state-action pairs have been collected by running some policy and $d_m(\boldsymbol{s}, \boldsymbol{a})$ is the empirical distribution on this dataset. If we collect an additional $k$ state-action pairs using the following distribution, and if $(m + k)d_{\pi_{\boldsymbol{\theta}}}(\boldsymbol{s}, \boldsymbol{a}) \geq m \cdot d_m(\boldsymbol{s}, \boldsymbol{a})$, then the aggregate empirical distribution over the $m + k$ pairs is equal to $d_{\pi_{\boldsymbol{\theta}}}(\boldsymbol{s}, \boldsymbol{a})$ in expectation:*

$$d_b(\boldsymbol{s}, \boldsymbol{a}) \coloneqq \frac{1}{Z}\left[d_{\pi_{\boldsymbol{\theta}}}(\boldsymbol{s}, \boldsymbol{a}) + \frac{m}{k}\left(d_{\pi_{\boldsymbol{\theta}}}(\boldsymbol{s}, \boldsymbol{a}) - d_m(\boldsymbol{s}, \boldsymbol{a})\right)\right]$$

*where $Z = \sum_{(\boldsymbol{s}, \boldsymbol{a}) \in \mathcal{S} \times \mathcal{A}}\left[d_{\pi_{\boldsymbol{\theta}}}(\boldsymbol{s}, \boldsymbol{a}) + \frac{m}{k}\left(d_{\pi_{\boldsymbol{\theta}}}(\boldsymbol{s}, \boldsymbol{a}) - d_m(\boldsymbol{s}, \boldsymbol{a})\right)\right]$ is a normalization coefficient.*

*Proof.* The proof is identical to the proof of Proposition 3, replacing $\pi_{\boldsymbol{\theta}}(\boldsymbol{a}), \pi_m(\boldsymbol{a})$, and $\pi_b(\boldsymbol{a})$ with $d_{\pi_{\boldsymbol{\theta}}}(\boldsymbol{s}, \boldsymbol{a}), d_m(\boldsymbol{s}, \boldsymbol{a})$, and $d_b(\boldsymbol{s}, \boldsymbol{a})$. $\square$

n practice, we cannot sample directly from the visitation distribution $d_b(\boldsymbol{s}, \boldsymbol{a})$ in Proposition 4 and instead approximate sampling from this distribution by sampling from its corresponding policy $\pi_b(\boldsymbol{a}|\boldsymbol{s}) = d_b(\boldsymbol{s}, \boldsymbol{a})/\sum_{\boldsymbol{a}' \in \mathcal{A}} d_b(\boldsymbol{s}', \boldsymbol{a}')$.

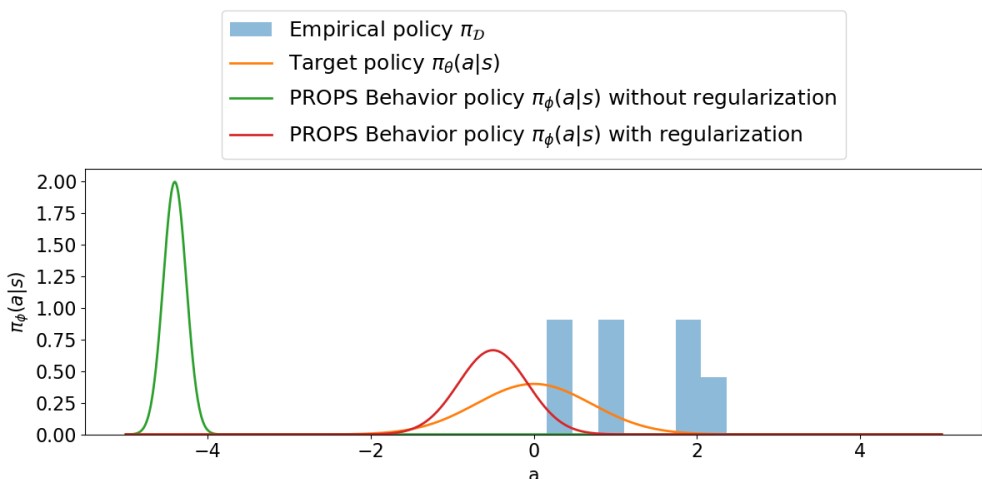

Figure 8: In this example, $\pi(\cdot|\boldsymbol{s}) = \mathcal{N}(0, 1)$. After several visits to $\boldsymbol{s}$, all sampled actions (blue) satisfy $a > 0$ so that actions $a < 0$ are under-sampled. Without regularization, PROPS will attempt to increase the probabilities of under-sampled actions in the tail of target policy distribution (green). The regularization term in the PROPS objective ensures the behavior policy remains close to target policy.

| $g(\boldsymbol{s}\,\boldsymbol{a}, \boldsymbol{\phi}, \boldsymbol{\theta}) > 0$ | Is the objective clipped? | Return value of min | Gradient |
|---|---|---|---|
| $g(\boldsymbol{s}\,\boldsymbol{a}, \boldsymbol{\phi}, \boldsymbol{\theta}) \in [1 - \epsilon_{\text{PROPS}}, 1 + \epsilon_{\text{PROPS}}]$ | No | $-g(\boldsymbol{s}, \boldsymbol{a}, \boldsymbol{\phi}, \boldsymbol{\theta})$ | $\nabla_{\boldsymbol{\phi}} \mathcal{L}_{\text{CLIP}}$ |
| $g(\boldsymbol{s}, \boldsymbol{a}, \boldsymbol{\phi}, \boldsymbol{\theta}) > 1 + \epsilon_{\text{PROPS}}$ | No | $-g(\boldsymbol{s}, \boldsymbol{a}, \boldsymbol{\phi}, \boldsymbol{\theta})$ | $\nabla_{\boldsymbol{\phi}} \mathcal{L}_{\text{CLIP}}$ |
| $g(\boldsymbol{s}, \boldsymbol{a}, \boldsymbol{\phi}, \boldsymbol{\theta}) < 1 - \epsilon_{\text{PROPS}}$ | Yes | $-(1 - \epsilon_{\text{PROPS}})$ | $\boldsymbol{0}$ |

Table 2: Behavior of PROPS's clipped surrogate objective (Eq. 8).

## C  PROPS Implementation Details

In this appendix, we describe two relevant implementation details for the PROPS update (Algorithm 2). We additionally summarize the behavior of PROPS's clipping mechanism in Table 2.

1. **PROPS update:** The PROPS update adapts the behavior policy to reduce sampling error in the buffer $\mathcal{D}$. When performing this update with a full buffer, we exclude the oldest batch of data collected by the behavior policy (*i.e.*, the $m$ oldest transitions in $\mathcal{D}$); this data will be evicted from the buffer before the next behavior policy update and thus does not contribute to sampling error in $\mathcal{D}$.

2. **Behavior policy class:** We compute behavior policies from the same policy class used for target policies. In particular, we consider Gaussian policies which output a mean $\mu(\boldsymbol{s})$ and a variance $\sigma^2(\boldsymbol{s})$ and then sample actions $\boldsymbol{a} \sim \pi(\cdot|\boldsymbol{s}) \equiv \mathcal{N}(\mu(\boldsymbol{s}), \sigma^2(\boldsymbol{s}))$. In principle, the target and behavior policy classes can be different. However, using the same class for both policies allows us to easily initialize the behavior policy equal to the target policy at the start of each update. This initialization is necessary to ensure the PROPS update increases the probability of sampling actions that are currently under-sampled with respect to the target policy.

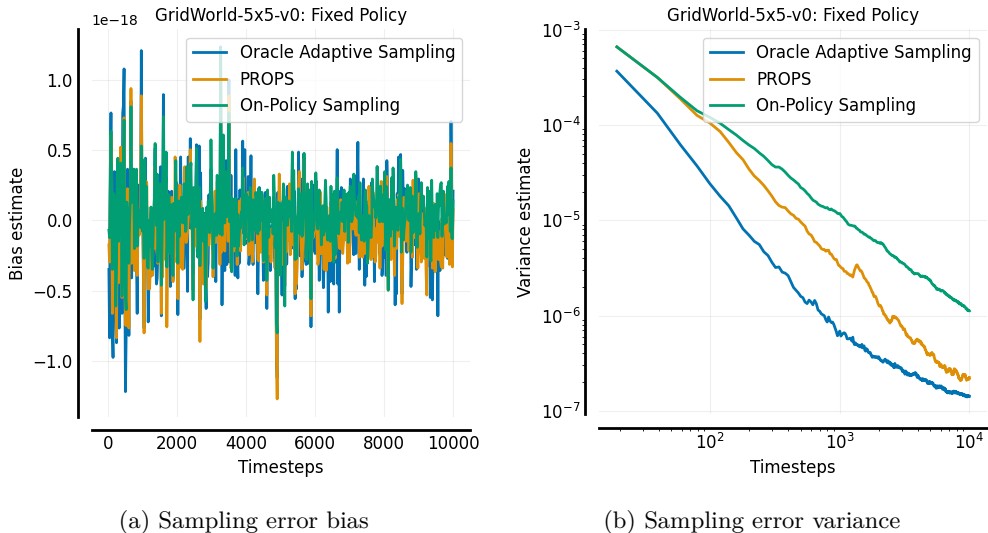

(a) Sampling error bias

(b) Sampling error variance

Figure 9: Sampling error bias and variance estimates of different sampling methods. Empirically, PROPS is unbiased and lower variance than on-policy sampling.

# D   Additional Experiments

In this appendix, we include additional experiments and ablations.

## D.1   Bias and Variance of PROPS

In Fig. 9, we investigate the bias and variance of the empirical state-action visitation distribution $d_{\mathcal{D}}(\boldsymbol{s}, \boldsymbol{a})$ under PROPS, ROS, and on-policy sampling. We report the bias and variance averaged over all $(\boldsymbol{s}, \boldsymbol{a}) \in \mathcal{S} \times \mathcal{A}$ computed as follows:

$$\text{bias} = \frac{1}{|\mathcal{S} \times \mathcal{A}|} \sum_{(\boldsymbol{s}, \boldsymbol{a}) \in \mathcal{S} \times \mathcal{A}} \left( \mathbb{E}\left[ d_{\mathcal{D}}(\boldsymbol{s}, \boldsymbol{a}) \right] - d_{\pi_{\boldsymbol{\theta}}}(\boldsymbol{s}, \boldsymbol{a}) \right) \tag{13}$$

$$\text{variance} = \frac{1}{|\mathcal{S} \times \mathcal{A}|} \sum_{(\boldsymbol{s}, \boldsymbol{a}) \in \mathcal{S} \times \mathcal{A}} \mathbb{E}\left[ \left( d_{\mathcal{D}}(\boldsymbol{s}, \boldsymbol{a}) - d_{\pi_{\boldsymbol{\theta}}}(\boldsymbol{s}, \boldsymbol{a}) \right)^2 \right] \tag{14}$$

As shown in Fig. 9, the visitation distributions under PROPS and ROS empirically have near zero bias (note that the vertical axis has scale $10^{-18}$) and have lower variance than on-policy sampling.

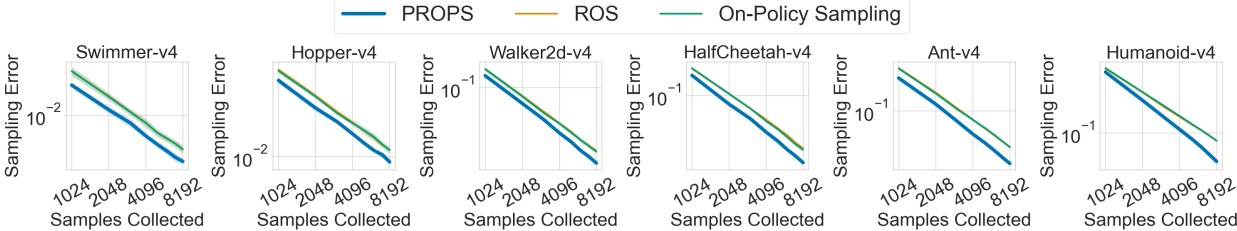

Figure 10: Sampling error with a fixed, expert target policy. Solid curves denote the mean over 5 seeds. Shaded regions denote 95% confidence belts.

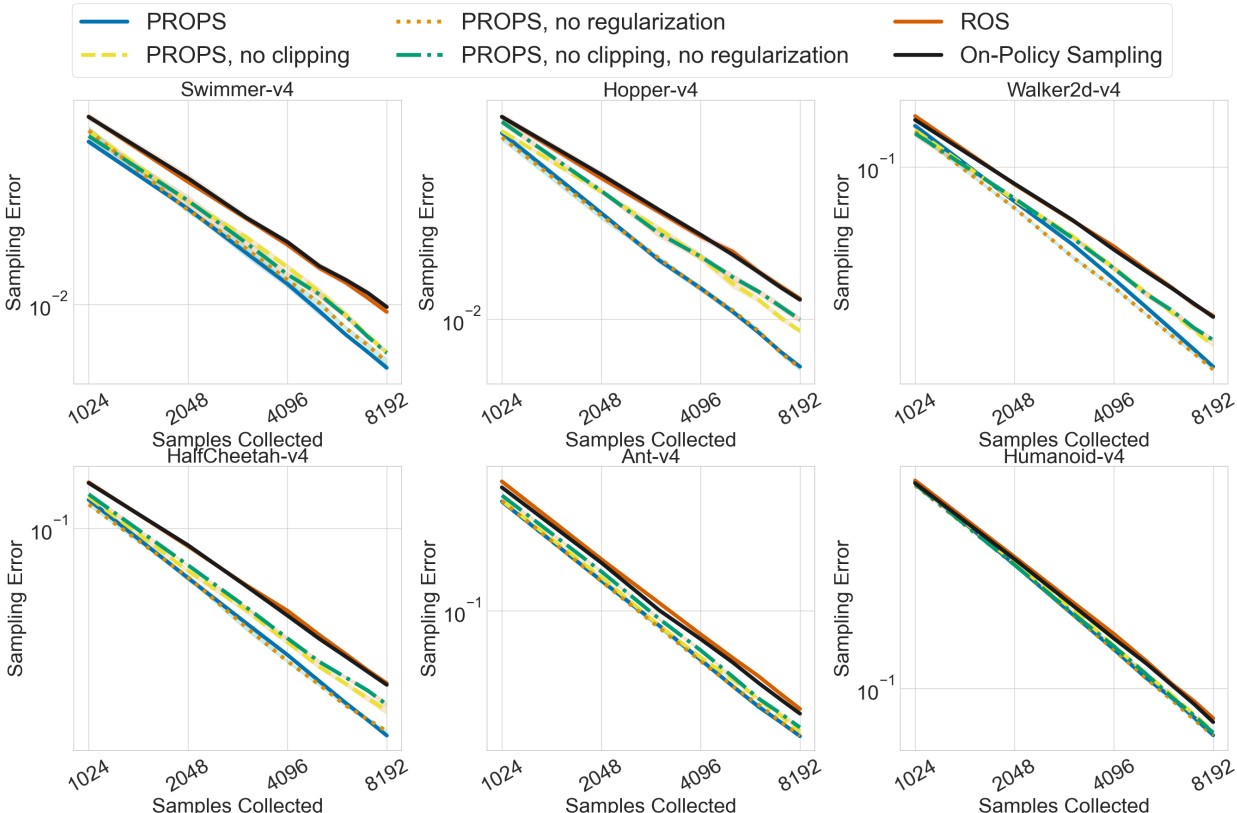

Figure 11: Sampling error ablations with a fixed, random target policy. Here, "no clipping" refers to setting $\epsilon_{\mathrm{PROPS}} = \infty$, and "no regularization" refers to setting $\lambda = 0$. Solid curves denote the mean over 10 seeds, and shaded regions denote 95% bootstrap confidence intervals.

## D.2 Correcting Sampling Error for a Fixed Target Policy

In this appendix, we expand upon results presented in Section 6.2 of the main paper and provide additional experiments investigating the degree to which PROPS reduces sampling error with respect to a fixed, randomly initialized target policy. We additionally include ablation studies investigating the effects of clipping and regularization.

We tune PROPS and ROS using a hyperparameter sweep. For PROPS, we sweep over learning rates in $\{10^{-3}, 10^{-4}\}$ and fix the remaining PROPS hyperparameters: regularization coefficient $\lambda = 0.1$, target KL $\delta_{\mathrm{PROPS}} = 0.03$, and clipping coefficient $\epsilon_{\mathrm{PROPS}} = 0.3$. For ROS, we sweep over learning rates in $\{10^{-3}, 10^{-4}, 10^{-5}\}$. We report results for hyperparameters yielding the lowest sampling error.

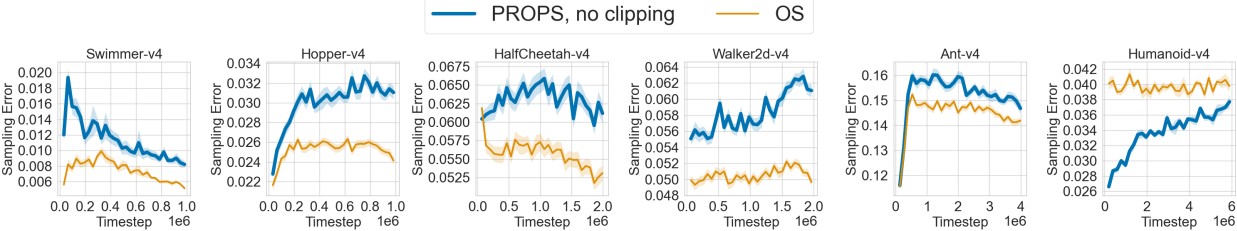

Figure 12: Sampling error throughout RL training without clipping the PROPS objective. Solid curves denote the mean over 5 seeds. Shaded regions denote 95% confidence belts.

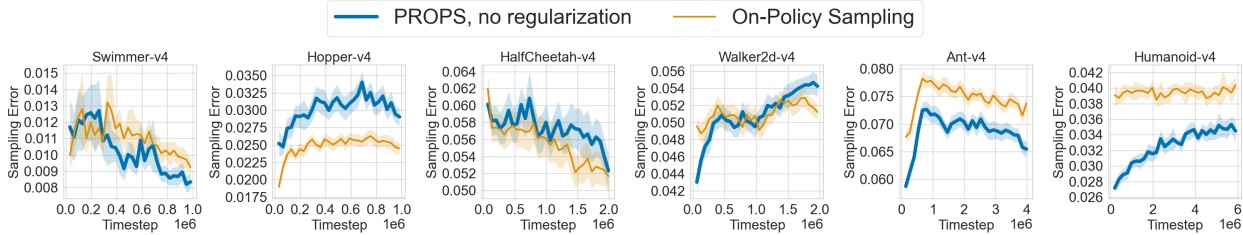

Figure 13: Sampling error throughout RL training without regularizing the PROPS objective. Solid curves denote the mean over 5 seeds. Shaded regions denote 95% confidence belts.

In Fig. 10, we see that PROPS achieves lower sampling error than both ROS and on-policy sampling across all tasks. ROS shows little to no improvement over on-policy sampling, again highlighting the difficulty of applying ROS to higher dimensional tasks with continuous actions.

Fig. 11 ablates the effects of PROPS's clipping mechanism and regularization on sampling error reduction. We ablate clipping by setting $\epsilon_{\mathrm{PROPS}} = \infty$, and we ablate regularization by setting $\lambda = 0$. We use a fixed expert target policy and use the same tuning procedure described earlier in this appendix. In all tasks, PROPS achieves higher sampling error without clipping or regularization than it does with clipping and regularization, though this method nevertheless outperforms on-policy sampling in all tasks. Only removing clipping increases sampling error in most setups, and only removing regularization often increases sampling error for smaller batches of data *e.g.*, 1024 samples. These observations indicate that while regularization is helpful, clipping has a stronger effect on sampling error reduction than regularization when the target policy is fixed.

### D.3 Correcting Sampling Error During RL Training

In this appendix, we include additional experiments investigating the degree to which PROPS reduces sampling error during RL training, expanding upon results presented in Section 6.3 of the main paper. We include sampling error curves for all six MuJoCo benchmark tasks and additionally provide ablation studies investigating the effects of clipping and regularization on sampling error reduction and data efficiency in the RL setting. We ablate clipping by tuning RL agents with $\epsilon_{\mathrm{PROPS}} = \infty$, and we ablate regularization by tuning RL agents with $\lambda = 0$. Fig. 12 and Fig. 13 show sampling error curves without clipping and without regularization, respectively. Without clipping, PROPS achieves larger sampling error than on-policy sampling in all tasks except Humanoid. Without regularization, PROPS achieves larger sampling error in 3 out of 6 tasks. These observations indicate that while clipping and regularization both help reduce sampling error during RL training, clipping has a stronger effect on sampling error reduction. As shown in Fig. 14 PROPS data efficiency generally decreases when we remove clipping or regularization.

Lastly, we consider training with larger buffer sizes $b$ in Fig. 15. We find that data efficiency may decrease with a larger buffer size. Intuitively, the more historic data kept around, the more data that must be collected to impact the aggregate data distribution.

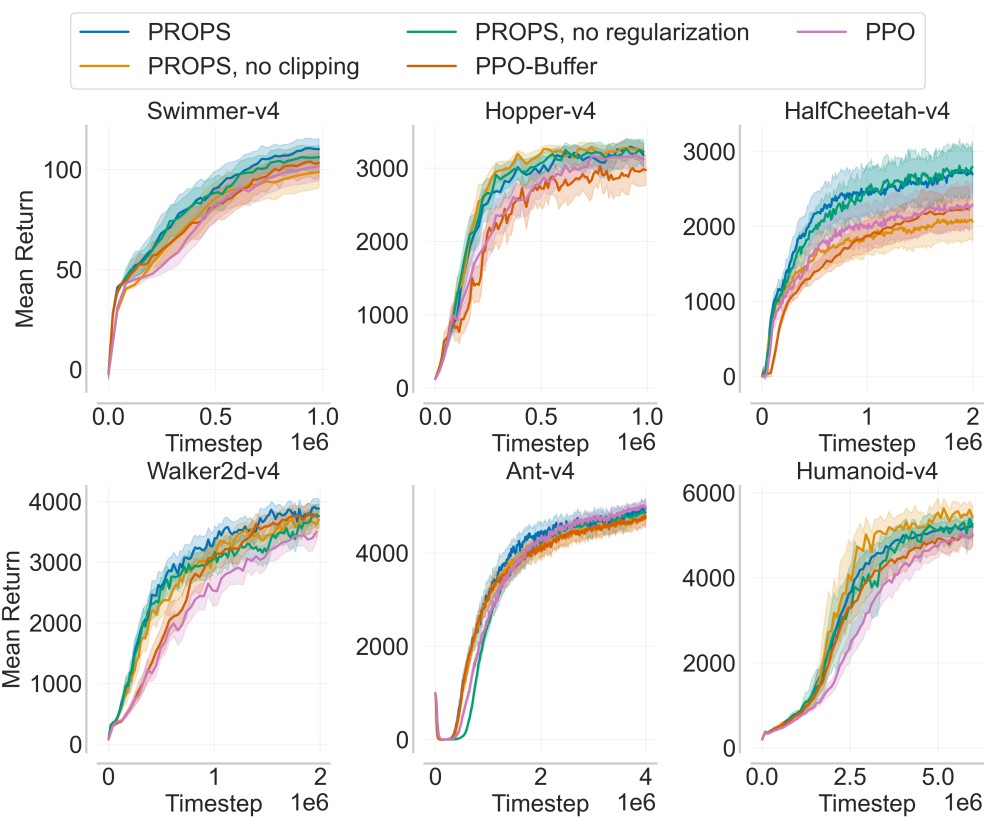

Figure 14: Mean return over 50 seeds of PROPS with and without clipping or regularizing the PROPS objective. Shaded regions denote 95% bootstrap confidence intervals.

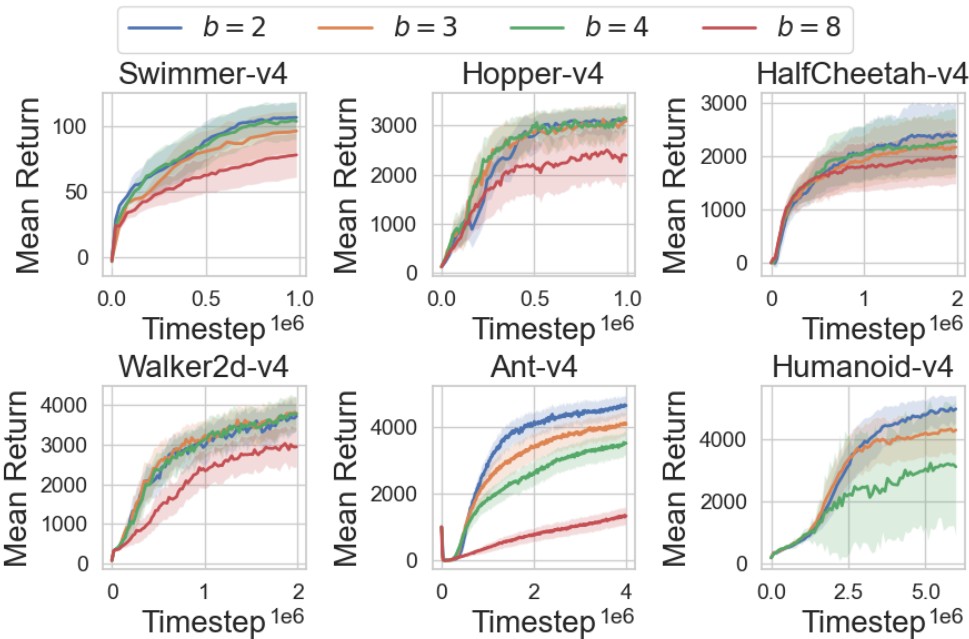

Figure 15: Mean return over 50 seeds for PROPS with different buffer sizes. We exclude $b = 8$ for Humanoid-v4 due to the expense of training and tuning. Shaded regions denote 95% bootstrap confidence intervals.

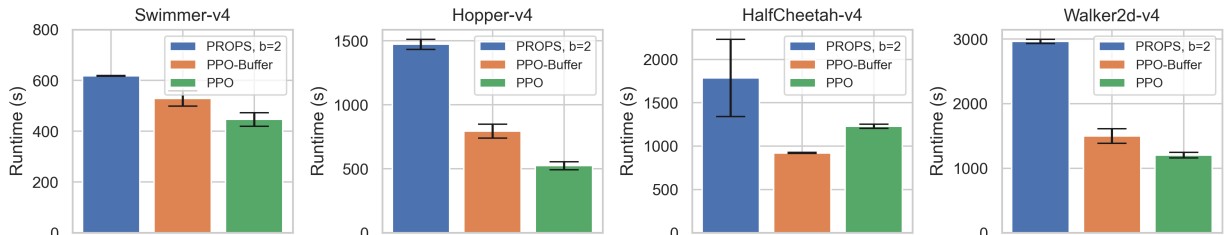

Figure 16: Runtimes for PROPS, PPO-Buffer, and PPO. We report means and standard errors over 3 independent runs.

| PPO learning rate | $10^{-3}, 10^{-4}$, linearly annealed to 0 over training |
|---|---|
| PPO batch size $n$ | $1024, 2048, 4096, 8192$ |
| PROPS learning rate | $10^{-3}, 10^{-4}$ (and $10^{-5}$ for Swimmer) |
| PROPS behavior update frequency $m$ | $256, 512, 1024$ |
| PROPS KL cutoff $\delta_{\text{PROPS}}$ | $0.03, 0.05, 0.1$ |
| PROPS regularizer coefficient $\lambda$ | $0.1, 0.3$ |

Table 3: Hyperparameters used in our hyperparameter sweep for RL training.

# E Runtime Comparisons

Figure 16 shows runtimes for PROPS, PPO-Buffer, and PPO averaged over 3 runs. We trained all agents on a MacBook Air with an M1 CPU and used the same tuned hyperparameters used throughout the paper. PROPS takes at most twice as long as PPO-Buffer; intuitively, both PROPS and PPO-Buffer learn from the same amount of data but PROPS learns two policies.

We note that PPO-Buffer is faster than PPO in HalfCheetah-v4 because, with our tuned hyperparameters, PPO-Buffer performs fewer target policy updates than PPO. In particular, PPO-Buffer is updating its target policy every 4096 steps, whereas PPO is updating the target policy every 1024 steps.

# F Hyperparameter Tuning for RL Training

For all RL experiments in Section 6.3 and Appendix D.3, we tune PROPS, PPO-Buffer, and PPO separately using a hyperparameter sweep over parameters listed in Table 3 and fix the hyperparameters in Table 5 across all experiments. Since we consider a wide range of hyperparameter values, we ran 10 independent training runs for each hyperparameter setting. We then performed 50 independent training runs for the hyperparameter settings yielding the largest returns at the end of RL training. We report results for these hyperparameters in the main paper.

| Environment | PPO Batch Size | PPO Learning Rate | PROPS Update Freq. | PROPS Learning Rate | PROPS KL Cutoff | PROPS Regularization $\lambda$ |
|---|---|---|---|---|---|---|
| Swimmer-v4 | 2048 | $10^{-3}$ | 1024 | $10^{-5}$ | 0.03 | 0.1 |
| Hopper-v4 | 2048 | $10^{-3}$ | 256 | $10^{-3}$ | 0.05 | 0.3 |
| HalfCheetah-v4 | 1024 | $10^{-4}$ | 512 | $10^{-3}$ | 0.05 | 0.3 |
| Walker2d-v4 | 2048 | $10^{-3}$ | 256 | $10^{-3}$ | 0.1 | 0.3 |
| Ant-v4 | 2048 | $10^{-4}$ | 256 | $10^{-3}$ | 0.03 | 0.1 |
| Humanoid-v4 | 8192 | $10^{-4}$ | 256 | $10^{-4}$ | 0.1 | 0.1 |

Table 4: Hyperparameters selected from our hyperparameter sweep for RL training.

| | |
|---|---|
| PPO number of update epochs | 10 |
| PROPS number of update epochs | 16 |
| Buffer size $b$ | 2 target batches (also 3, 4, and 8 in Fig. 15) |
| PPO minibatch size for PPO update | $bn/16$ |
| PROPS minibatch size for PROPS update | $bn/16$ |
| PPO and PROPS networks | Multi-layer perceptron |
| | with hidden layers (64,64) |
| PPO and PROPS optimizers | Adam (Kingma & Ba, 2015) |
| PPO discount factor $\gamma$ | 0.99 |
| PPO generalized advantage estimation (GAE) | 0.95 |
| PPO advantage normalization | Yes |
| PPO loss clip coefficient | 0.2 |
| PPO entropy coefficient | 0.01 |
| PPO value function coefficient | 0.5 |
| PPO and PROPS gradient clipping (max gradient norm) | 0.5 |
| PPO KL cut-off | 0.03 |
| Evaluation frequency | Every 10 target policy updates |
| Number of evaluation episodes | 20 |

Table 5: Hyperparameters fixed across all experiments. We use the PPO implementation provided by CleanRL (Huang et al., 2022).

## G Hyperparameter Sensitivity Analysis

To assess how sensitive PROPS is to hyperparameter selection, we follow Patterson et al. (2023) and plot RL training performance aggregated over a large subset of runs from our hyperparameter sweep. Although we include the PPO target learning rate and PPO target batch size in our hyperparameter sweep, we keep them fixed at their tuned values in our violin plots. Including these hyperparameters would tell us about the sensitivity of PPO, but we are only interested in understanding the sensitivity of PROPS. We run 10 seeds for each hyperparameter setting. Fig. 17 and Fig. 18 show training curves and performance profiles for tuned PROPS and aggregated PROPS as well as our baselines. Across tasks, tuned PROPS achieves higher returns than aggregated PROPS, demonstrating the benefit of careful hyperparameter tuning. Nevertheless, aggregated PROPS still outperforms PPO and PPO-Buffer, suggesting that PROPS maintains a degree of hyperparameter robustness.

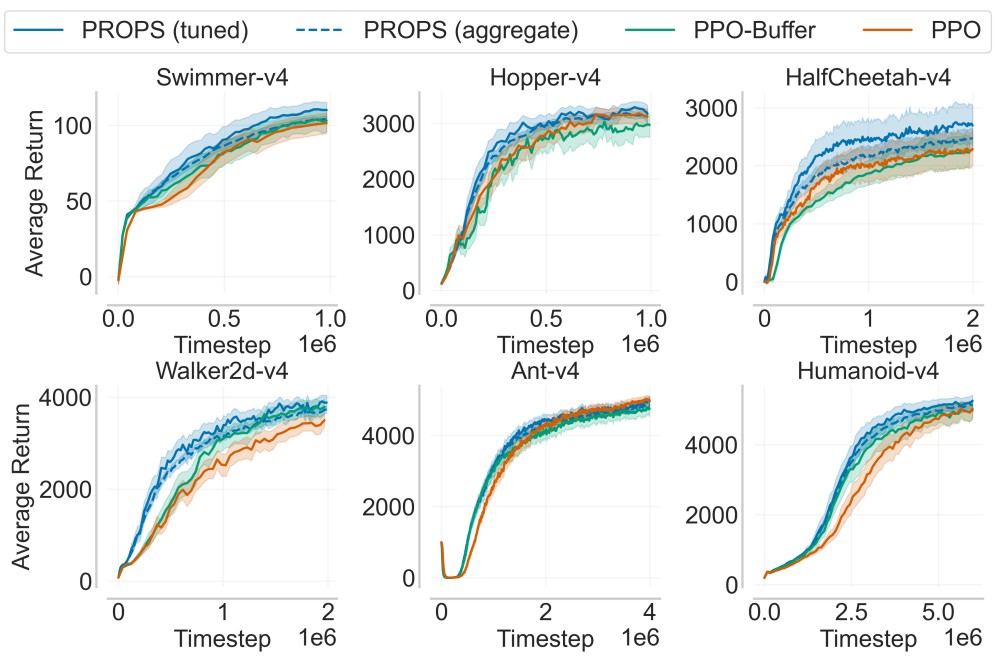

Figure 17: Mean return over 50 seeds. Shaded regions denote 95% confidence intervals.

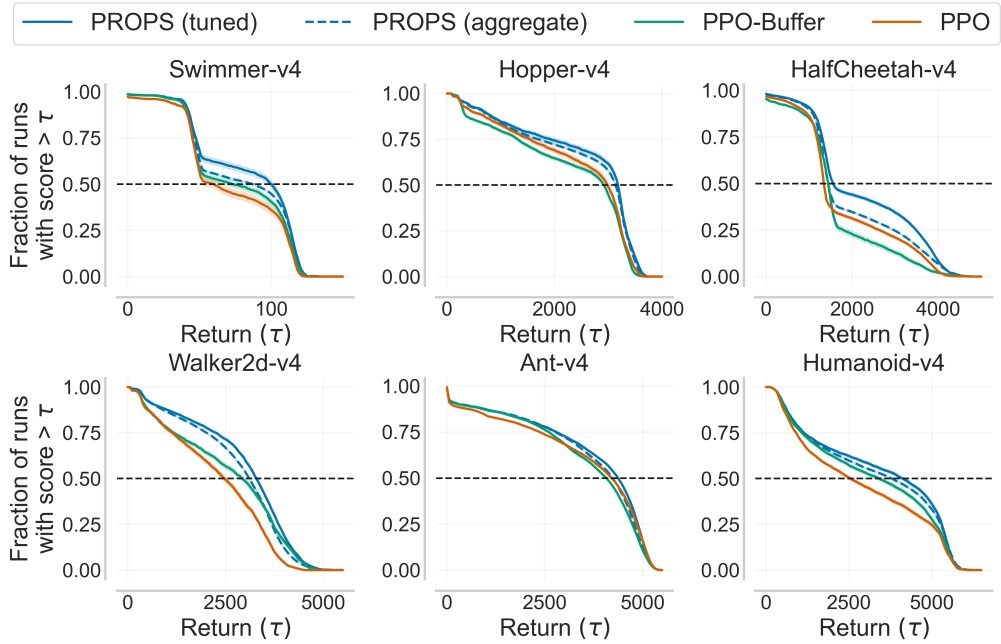

Figure 18: Performance profiles over 50 seeds. Shaded regions denote 95% confidence intervals.

