# OpenReview forum: "On-Policy Policy Gradient Reinforcement Learning Without On-Policy Sampling"
_TMLR — Accepted by TMLR_

### Review · Reviewer_gaQF · 2025-11-16

**Summary Of Contributions:**

This paper investigates the on-policy policy gradient RL without on-policy sampling. In particular, the paper builds upon [A1], which shows that instead of on-policy evaluation, one can use the existing method with some carefully crafted behavioral policy (ROS), which can result in sampling efficiency.  The main contributions over [A1] are that the work is only related to a finite action space; here, the paper considers a continuous action space. In particular, the paper proposes a KL regularization objective and PPO PPO-based technique to ensure that (i) the behavioral policy should not reweight the updates using the data that might have been obtained using a very old policy, and (ii) the behavioral policy does not move much from the target policy. The paper has conducted extensive empirical evaluations to validate its proposed approach.

Strengths:
1. The paper has explained the approaches and the rationale well enough.

2. The empirical evidence suggests that the proposed approach indeed has an advantage (even though minute) over the ROS.

Weaknesses:

1. Even though the empirical results show improvements, they are very minute.

2. The novelties are also very limited. In particular, it is eventually the ROS with some KL regularization and PPO-based clipping. Neither is novel to the RL.

 3. In Figure 5, the sampling error fluctuate a lot, hence, it might be disputed to conclude that it has a better advantage over the on-policy sampling.

[A1].Rujie Zhong, Duohan Zhang, Lukas Schäfer, Stefano Albrecht, and Josiah Hanna. Robust on-policy sampling
for data-efficient policy evaluation in reinforcement learning. Advances in Neural Information Processing
Systems, 35:37376–37388, 2022.

**Additional Comments:**

1. As a metric, the authors used sampling error. I would suggest defining this formally, so that everyone can understand the implications.

2. Can the authors comment on the theoretical properties? I mean can the authors show the convergence, and sampling efficiency with respect to RPO in any manner?

**Audience:**

Yes

**Audience Explanation:**

This paper would be of interest to the RL community in general.

**Claims And Evidence:**

No

**Claims Explanation:**

While the empirical evidence suggests some improvement for a continuous action space. However, it is not clear whether the proposed approach can have a significant advantage over the existing approaches. Also, for some tasks, the sampling error is increased. Further, the paper did not make any comparisons with other off-policy algorithms.

**Requested Changes:**

1. The paper provides a lot of examples of why on-policy sampling might not be a good idea for policy evaluation; one needs to use the existing data. In my opinion, it is unnecessary, as we have a pretty good idea from Zhong et al. Rather, I would suggest moving some of the theoretical results and explanations to the main text.

2. The algorithm is similar to the two-time-scale approach as the target distribution is updated in the outer loop, and the behavioral policy is updated in the inner loop. Can the authors comment on the learning rates for both these loops?

3. I would suggest comparing this approach with off-policy evaluation approaches (E.g. [A2])

[A2]. Liu, Shuze Daniel, Claire Chen, and Shangtong Zhang. "Doubly optimal policy evaluation for reinforcement learning." arXiv preprint arXiv:2410.02226 (2024). https://arxiv.org/pdf/2410.02226

---

> ### Author Response · Authors · 2025-12-10
>
> Thank you for the thoughtful review! We are glad that you think this work would be of interest to the RL community. Below, we clarify the exact claims we make in our work and elaborate on how the existing experimental and theoretical results support these claims.
>
> > Even though the empirical results show improvements, they are very minute.
>
> The central claim we make is that PROPS reduces sampling error more than on-policy sampling and, as a result, improves data efficiency. Our experiments support this claim: Figure 5 shows that PROPS reduces sampling error during RL training and Figure 4a shows that PROPS achieves higher return with fewer environment interactions than both PPO and PPO-Buffer. Figure 4b demonstrates that PROPS shifts the distribution of final returns toward larger values, increasing the fraction of runs that achieve higher performance compared to the baselines.
>
> > The novelties are also very limited. In particular, it is eventually the ROS with some KL regularization and PPO-based clipping. Neither is novel to the RL.
>
> Our paper does not claim novelty in KL regularization, PPO-style clipping, or the concept of using adaptive sampling to reduce sampling. Instead, our contribution is to show that integrating KL regularization and PPO-style clipping is needed to scale the ROS sampling-error correction method from low-dimensional **policy evaluation** settings to higher-dimensional continuous-action **reinforcement learning**, where ROS on its own fails.
>
> >  In Figure 5, the sampling error fluctuate a lot, hence, it might be disputed to conclude that it has a better advantage over the on-policy sampling.
>
> In our revisions, Figure 5 now shows sampling error averaged over 10 seeds. Results are qualitatively the same: In 5 of the 6 tasks, PROPS achieves lower or similar sampling error than on-policy sampling at all points during training.
>
> >the paper did not make any comparisons with other off-policy algorithms
>
> We do not compare to off-policy algorithms because PROPS is intended to improve the data efficiency of on-policy policy-gradient methods relative to on-policy sampling. We do not claim that PROPS makes PPO competitive with fully off-policy RL, only that it improves efficiency within the on-policy setting. Although PROPS can reuse historic data, it should be viewed as an enhancement to on-policy learning, not an off-policy alternative.
>
> > The paper provides a lot of examples of why on-policy sampling might not be a good idea for policy evaluation; one needs to use the existing data. In my opinion, it is unnecessary, as we have a pretty good idea from Zhong et al.
>
> We agree that Zhong et al. already provide a useful overview of sampling error, but we include our own discussion to keep the paper self-contained and to highlight a distinct point: Zhong et al. study sampling error in **policy evaluation**, whereas our focus is on how sampling error affects **policy-gradient RL**. In particular, Zhong et al. examine how sampling error affects **value function estimates**, whereas we focus on how sampling error affects **policy-gradient estimates** and can ultimately lead to suboptimal convergence.
>
> > The algorithm is similar to the two-time-scale approach as the target distribution is updated in the outer loop, and the behavioral policy is updated in the inner loop. Can the authors comment on the learning rates for both these loops?
>
> All hyperparameters, including learning rates for the target and behavior policies, are in Table 3 of our submission. Behavior policy learning rates are the same as the target policy learning rate in many tasks.
>
> >  I would suggest comparing this approach with off-policy evaluation approaches
>
> Thank you for pointing us to this reference; We agree that it’s relevant and now include it in our related-work section. However, we believe a comparison to Doubly Optimal (DOpt) Policy Evaluation is not appropriate because that method is designed for **offline policy evaluation**, whereas PROPS focuses on **online policy-gradient RL**. Extending DOpt to online RL may be a separate paper on its own---indeed, the authors note this as intended future work..
>
> A practical challenge is that DOpt assumes access to a large, diverse offline dataset (the authors use 1000 trajectories in their experiments); the authors explicitly acknowledge this data requirement as a key limitation of DOpt and suggest using on-policy data collection if such an offline dataset is not available. In on-policy RL, the data buffer is comparatively small (often just a few trajectories) and approximately distributed on-policy, so it’s unclear if DOpt can be applied out-of-the-box to on-policy RL without modification.
>
> A second, minor point is that DOpt’s MuJoCo experiments truncate trajectories to 100 steps, while the standard truncation horizon is 1000. It is unclear if DOpt scales to longer-horizons typical in RL tasks.

---

> ### Author Response · Authors · 2025-12-10
>
> > As a metric, the authors used sampling error. I would suggest defining this formally, so that everyone can understand the implications.
>
> In our revisions, we moved the description of the sampling error metric from the appendix to the experiments section (Section 6.1).
>
> > Can the authors comment on the theoretical properties? I mean can the authors show the convergence, and sampling efficiency with respect to RPO in any manner?
>
> Could you please clarify what RPO refers to? Our understanding is that RPO refers to this paper: https://arxiv.org/abs/2212.07536.
>
> Our work focuses on scaling the ideas from ROS to continuous control **empirically.** The theoretical result we provide is in the tabular setting and is intended as conceptual grounding for PROPS. Extending the analysis of Zhong et al. [2] to PROPS in continuous-control settings would be valuable future work, and we note this in the Discussion section.
>
>
> Thank you again for the useful suggestions! We hope our response clarifies how our work supports our stated claims. Please let us know if there are further questions; we are happy to discuss!
>
> ---
> 1. Zhong et al. Robust on-policy sampling for data-efficient policy evaluation in reinforcement learning. NeurIPS 2022.
> 2. Liu et. al. Doubly Optimal Policy Evaluation for Reinforcement Learning. ICLR, 2025.

---

> > ### Comment · Reviewer_gaQF · 2025-12-23
> > **Thanks for the clarification**
> >
> > I would like to thank the authors for their efforts and clarifying all my doubts. However, my main concern remains. The improvement is very minute. Can you put some numbers or a table in terms of how much improvement you are getting?
> > In particular, I want to understand how you are proving your claim that ``integrating KL regularization and PPO-style clipping is needed to scale the ROS sampling-error correction method from low-dimensional policy evaluation settings to higher-dimensional continuous-action reinforcement learning, where ROS on its own fails."

---

> ### Author Response · Authors · 2026-01-04
>
> Thank you for the follow-up comment! We have updated our submission and highlighted our latest revisions in brown. Below, we (1) clarify our stated contributions and (2) explain why our empirical results demonstrate meaningful improvements in data efficiency.
>
> > I want to understand how you are proving your claim that ``integrating KL regularization and PPO-style clipping is needed to scale the ROS sampling-error correction method from low-dimensional policy evaluation settings to higher-dimensional continuous-action reinforcement learning, where ROS on its own fails."
>
> Our original use of the phrase “is needed to scale” is a bit ambiguous, so we clarify our intended meaning here. This claim restates Contribution 2 listed at the end of our introduction: PROPS (*which can be viewed as ROS with KL regularization and PPO-style clipping*) reduces sampling error at a faster rate than on-policy sampling in continuous tasks, whereas ROS on its own does not. **Since batch sizes in the range of 2048–8192 are standard for RL in these tasks, ROS alone will not be able to improve data efficiency.** This claim is primarily supported by Figure 3: ROS does not reduce sampling error any more than on-policy sampling within 8192 interactions on MuJoCo tasks. Ablations in Figures 12 and 13 of Appendix E.3 also show that both clipping and KL regularization are needed to reduce sampling error during RL training.
>
> Our second claim (Contribution 3) is that reducing sampling error via PROPS increases data efficiency in on-policy policy-gradient RL. In our revisions, we now clarify that **"improving data efficiency" means reducing the number of environment interactions required to achieve returns comparable to those obtained with on-policy sampling.** In other words, we claim that PROPS improves learning speed, not necessarily final performance at the end of training.
>
> To better illustrate the practical impact of PROPS, we updated the submission with two additional summaries: Table 1 and Figure 5 (pages 10–11).
>
> * Table 1 reports average returns for PROPS, PPO-Buffer, and PPO at 33%, 66%, and 100% of the training budget. We highlight improvements that are statistically significant according to a paired t-test at a 95% confidence level. These results show that PROPS reaches returns comparable to PPO earlier in training. For instance, in Humanoid-v4 and Hopper-v4, PROPS closely matches the final return of PPO  with 33% fewer interactions.
> * To provide a more compact and readable summary of performance throughout training, we also report results aggregated across all six MuJoCo tasks in Fig. 5. PROPS matches the normalized return of PPO and PPO-Buffer using only 60% of the training budget. Non-overlapping 95% bootstrap confidence intervals over 20% to 60% of the training budget provide a conservative visual indication of statistical significance. Figure 5b. shows the aggregate performance profiles, showing that PROPS shifts the aggregate return distribution toward larger values.
>
> Here is a summary of our newest revisions:
> 1. `Section 6`: Clarify what we mean by “data efficiency” in Q2.
> 2. `Section 6.2`: Clarify that ROS does not scale to RL because it does not reduce sampling error after collecting 8192 samples (a standard batch size for RL on the MuJoCo benchmark)
> 3. `Section 6.3`: Table 1, Figure 5, and in-text descriptions of their results.
>
> We hope this clarifies our claim about clipping and KL regularization as well as why our results show meaningful improvements in data efficiency. We are happy to further clarify or refine the presentation based on further suggestions if you have any!

---

> > ### Comment · Reviewer_gaQF · 2026-01-07
> > **Thank you**
> >
> > I am now fine with accepting this paper.

---

### Review · Reviewer_JSKr · 2025-11-18

**Summary Of Contributions:**

The paper develops the idea of using off-policy data collection to obtain an on-policy data distribution in order to improve the Monte-Carlo estimate in on-policy policy gradient methods, an idea initially formalized in [1]. This correction improves the gradient estimates and, in turn, the efficiency and performance of on-policy policy gradient methods. The key contribution is to scale this approach to high-dimensional, continuous problems. To achieve this goal, the authors
- remove the strong DAG assumption from [1] in the theoretical results for discrete state and action spaces.
- introduce PPO-like gradient clipping to the objective introduced in [1]
- add a KL regularization between behavior and target policy to the objective introduced in [1]

Strengths and Weaknesses:

S1: The paper makes a valuable contribution by scaling a recent sampling error correction method to standard deep RL benchmark problems by modifying the objective function with commonly used clipping and regularization terms.

S2: The paper is well written and discusses findings and limitations in a transparent manner.

W1: The theoretical reasoning heavily builds on [1] and solely considers finite-horizon MDPs with discrete state and action spaces, while the method is applied to infinite-horizon MDPs with continuous state and action spaces. From my understanding, scaling the approach to these more challenging settings is the key contribution of the paper. Thus, theoretical reasoning should be adapted to this type of MDP.

[1] Zhong et al., Robust On-Policy Sampling for Data-Efficient Policy Evaluation in Reinforcement Learning, NeurIPS 2022

**Audience:**

Yes

**Audience Explanation:**

On-policy policy gradient methods are of major importance in RL research and are probably the most common RL approach in applied fields like robotics. Improving the stability and efficiency of these methods is of interest to a large audience.

**Broader Impact Concerns:**

-

**Claims And Evidence:**

Yes

**Claims Explanation:**

Theoretical results:
All theoretical results discuss finite state and action spaces and finite-horizon MDPs. The method, however, is evaluated in MuJoCo locomotion tasks, which have continuous state and action spaces and, to the best of my knowledge, are formulated as infinite-horizon MDPs (truncation rather than termination at the end of an episode). The authors discuss in the limitations section that they do not focus on the theoretical properties of the method. Nonetheless, I think the technical reasoning is rather weak, given that the key contribution is scaling findings from [1] to continuous spaces. While it is probably hard to give the same convergence guarantees as in the tabular setting, I would have expected some reasoning on whether findings of [1] carry over to continuous state and action spaces (and infinite horizon MDPs) and what potential problems might be.

Empirical results:
The empirical results are mostly convincing. A clear reduction in sampling error, as well as moderate efficiency and performance gains, can be observed. Some experiments are solely run with 5 random seeds (e.g., Fig. 3, 5). In those cases, it would be preferable to have at least 10 seeds.
A limitation is the fact that the influence of the off-policy data generation on the advantage estimate is not considered. This is made clear by the authors, and I assume it is unlikely that the efficiency and performance gains are due to some effect on the advantage estimation rather than the sampling error correction. It is not trivial to find an ablation where it is possible to show this definitively, so I think the experiments can remain unchanged. However, if the authors have an idea how to do such an ablation, the paper would benefit from it.

[1] Zhong et al., Robust On-Policy Sampling for Data-Efficient Policy Evaluation in Reinforcement Learning, NeurIPS 2022

**Requested Changes:**

C1: The finite MDP setting seems a bit limiting. As far as I know, the MuJoCo locomotion tasks you benchmark on are infinite-horizon tasks as they truncate rather than terminate trajectories at the end of an episode, so at least empirically, your method seems to work in the infinite-horizon setting. Please elaborate on why the finite horizon assumption is necessary or provide results for infinite horizon MDPs.

C2: Please discuss a bit more how and if results from [1] are applicable to continuous state and action spaces. I understand that this paper is more application-oriented than [1] and that an in-depth analysis of continuous spaces is not trivial. Nonetheless, it would be preferable to have a clear distinction between results that are easily transferable to continuous spaces and findings that are hard to prove in the continuous setting. In the latter case, it is ok to rely on intuition from the tabular setting, as long as this is made transparent.

C3: Please discuss the computational overhead of your method in the empirical results. E.g., discuss the average run time of PROPS vs. PPO on your hardware.

Minor:

C4: Please make sure you have at least 10 random seeds for all experiments.

C5: (Section 3.1) It is a bit surprising that you introduce discrete state and action spaces since your contribution is scaling the method to continuous spaces. I would prefer the more general continuous case, which can be easily done by replacing the probability mass functions (p, \pi, d) with probability density functions.

C6: (Section 4) Please introduce the acronym DAG (I assume directed acyclic graph).

C7 : (Section 5) Before reading [1], it was not immediately clear to me why the loss function for adapting to the collection policy looks this way. To make the paper more self-contained, it would be good to explain that a bit more.

[1] Zhong et al., Robust On-Policy Sampling for Data-Efficient Policy Evaluation in Reinforcement Learning, NeurIPS 2022

---

> ### Author Response · Authors · 2025-12-10
>
> Thank you for the helpful feedback! We're pleased to see that you agree our work is a valuable contribution to the RL community and that found our submission well-written. Below, we address your comments. We've made minor revisions to our manuscript to address your comments.
>
> > A limitation is the fact that the influence of the off-policy data generation on the advantage estimate is not considered. This is made clear by the authors, and I assume it is unlikely that the efficiency and performance gains are due to some effect on the advantage estimation rather than the sampling error correction...if the authors have an idea how to do such an ablation, the paper would benefit from it.
>
> We agree that isolating the influence of off-policy data on the advantage estimate is important and technically challenging. The cleanest ablation would compute the exact advantage under the current policy for every (s,a) in the buffer (including those collected by historic policies) for PPO, PPO-Buffer, and PROPS. This would eliminate any off-policy advantage bias and isolate the effect of sampling error alone, but exact advantages are unfortunately difficult to compute in MuJoCo tasks.
>
> Nevertheless, experiments suggest that the effect of advantage bias is minimal. In our MuJoCo experiments, PPO-Buffer and PROPS both reuse historic data and use the same advantage estimator, so both methods have similar potential off-policy advantage bias. The main difference is that PROPS corrects sampling error in the buffer while PPO-Buffer does not. PROPS’s consistent improvement in data efficiency over PPO-Buffer therefore suggests that the gains stem from sampling error reduction rather than advantage bias.
>
> > Please elaborate on why the finite horizon assumption is necessary or provide results for infinite horizon MDPs.
>
> Thank you for raising this point. We specify finite-horizon MDPs in our preliminaries section to reflect standard modeling conventions in RL: although MuJoCo locomotion tasks are formally infinite-horizon, it is common practice to truncate trajectories after a fixed number of steps and reset the agent. PROPS makes no assumptions on finite vs. infinite horizon and can be used in both settings; the expected on-policy visitation distribution is different in either setting, but in either case, PROPS will track the on-policy distribution. To avoid confusion, we now specify infinite horizon MDPs in the Preliminaries section to better align with our experiments.
>
> The finite-horizon assumption appears in the ROS paper because their original convergence result relies on converting a finite-horizon MDP into a DAG. Our theoretical result removes this DAG restriction, so we no longer require the MDP to be finite-horizon.
>
> > Please discuss a bit more how and if results from [1] are applicable to continuous state and action spaces. I understand that this paper is more application-oriented than [1] and that an in- depth analysis of continuous spaces is not trivial. Nonetheless, it would be preferable to have a clear distinction between results that are easily transferable to continuous spaces and findings that are hard to prove in the continuous setting. In the latter case, it is ok to rely on intuition from the tabular setting, as long as this is made transparent.
>
> This is a good point which we now clarify in our revisions.  Our work focuses on scaling the ideas from ROS to continuous control **empirically**. Our theoretical contribution focuses on the tabular setting, and we agree that it would be useful but nontrivial to extend the analysis of Zhong et al. [1] to continuous state/action spaces. Our results do not immediately transfer to continuous control.
>
> What *does* transfer from [1] is the key intuition: sampling error can be reduced by increasing the probability of under-sampled actions via gradient updates on the negative log-likelihood. To scale this idea to higher-dimensional continuous control tasks, PROPS wraps this idea in a PPO-style update with KL regularization.
>
> > C3: Please discuss the computational overhead of your method in the empirical results. E.g., discuss the average run time of PROPS vs. PPO on your hardware.
>
> We have added a discussion of computational overhead in Appendix E. Empirically, PPO+PROPS requires at most 2x the wall-clock time of PPO with on-policy sampling on our hardware. This overhead is expected: PROPS learns a behavior policy in addition to the target policy, so the total number of policy-network updates roughly doubles. The actual overhead can be lower in practice, since the PROPS update terminates early once the behavior policy's KL reaches the chosen threshold, *i.e.* a smaller KL cutoffs could yield fewer behavior-policy updates.

---

> > ### Author Response · Authors · 2025-12-10
> >
> > > Please make sure you have at least 10 random seeds for all experiments.
> >
> > In our revisions, all sampling error curves now show 10 seeds. Results remain qualitatively the same.
> >
> > >  It is a bit surprising that you introduce discrete state and action spaces since your contribution is scaling the method to continuous spaces. I would prefer the more general continuous case, which can be easily done by replacing the probability mass functions (p, \pi, d) with probability density functions.
> >
> > We include both discrete and continuous state/action spaces in Section 3.1 because PROPS applies to both settings. We now note this point in our revision. We use discrete examples in Section 4 for ease of exposition; the intuition behind sampling error, its effect on policy gradient estimates, and how to reduce sampling error is simplest to illustrate in discrete spaces. When introducing the KL regularization used in PROPS, we then shift to continuous-action settings, since this modification is specifically needed to stabilize sampling-error correction in continuous control.
> >
> > > Please introduce the acronym DAG (I assume directed acyclic graph).
> >
> > We have fixed this in our revisions. Thank you for pointing this out!
> >
> > > Before reading [1], it was not immediately clear to me why the loss function for adapting to the collection policy looks this way. To make the paper more self-contained, it would be good to explain that a bit more.
> >
> > Thank you for raising this point! We have added additional exposition on the ROS method in Section 5.1. Please let us know if this addition clarifies how ROS works and how PROPS is connected to ROS.
> >
> > ---
> >
> > Thank you again for the useful suggestions! We hope our response addresses your comments. Please let us know if there are further questions; we are happy to discuss!
> >
> > 1. Zhong et al. Robust on-policy sampling for data-efficient policy evaluation in reinforcement learning. NeurIPS 2022.

---

> > > ### Comment · Reviewer_JSKr · 2025-12-12
> > >
> > > The revised version of the paper appropriately addresses my prior concerns.

---

### Review · Reviewer_a4R9 · 2025-11-26

**Summary Of Contributions:**

The paper builds on the Robust On-Policy Sampling (ROS) algorithm to reduce variance in the sampling process and obtain more accurate policy gradient estimates. The proposed method, PROPS, incorporates several techniques used in PPO to stabilize the ROS updates. This additional stability is shown to be beneficial in gridworld and continuous control benchmark tasks by reducing sampling error and improving policy optimization.
Ablations show the necessity of the additions to ROS.
Additionally, a new theoretical result is proved, generalizing one for the original ROS paper.

**Audience:**

Yes

**Audience Explanation:**

The proposed algorithm is a generic method to reduce sampling error when one can control the sampling distribution. While the core of the algorithm consists of the original ROS method, PROPS showcases that the method can work in more complex environments with some additional stabilizing techniques.
As such, I think this work would of interest to some members of the community, potentially beyond the RL sphere.

**Claims And Evidence:**

Yes

**Claims Explanation:**

The claims concerning the purported benefits of PROPS to reduce sampling error and its advantages in continuous control tasks over the original ROS algorithm are supported by proper experiments in a few environments.
While the diversity of environments is fairly limited, the experimental methodology seems sound with sufficient number of seeds and appropriate ablations.

I think the paper could benefit from some additional exposition for the ROS method since PROPS builds on it directly. It was initially unclear how the ROS gradient estimate is useful for addressing the undersampling/oversampling issue.

**Requested Changes:**

As mentioned above, it would help if there was a more in-depth description of ROS to make the paper clearer.

Minor changes/questions:
- Fig. 2, what's the oracle smallest variance possible with a given number of samples?

- Fig. 10, could the orange lines be changed to some different color? It's currently hard to tell the difference.

-  The text refers to Fig. 6b. and mentions  "5 out of 6 tasks" but the figure only shows GridWorld.

- In the paragraph under equation (7), it is mentioned that "We stop the PROPS update
early when $D_{KL}(\pi_\theat||\pi_\phi)$ reaches a chosen threshold $\delta{PROPS}$". Since there is also a KL regularizer between the two policies, are both of these methods necessary? It seems redundant.

---

> ### Author Response · Authors · 2025-12-10
>
> Thank you for the thoughtful feedback and suggestions! We're pleased to see that you agree our work is empirically sound and of interest to the RL community. We've made minor revisions to our manuscript in response to your comments, which we now address below.
>
> > it would help if there was a more in-depth description of ROS to make the paper clearer.
>
> Thank you for raising this point! We have added additional exposition on the ROS method in Section 5.1. Please let us know if this addition clarifies how ROS works and how PROPS is connected to ROS.
>
> > what's the oracle smallest variance possible with a given number of samples?
>
> This is a good question. The smallest possible sampling error depends on both the number of samples and the agent’s current policy probabilities and is difficult to compute in closed form. For example, if the agent’s policy places equal probability on three actions at some state,, zero sampling error is attainable with three samples (i.e. when the agent samples each action ones), but unattainable with two samples (one action is necessarily under-sampled) or four samples (one action is necessarily over-sampled), since the empirical frequencies cannot exactly match the target distribution in those cases. In general, as the number of samples increases, the minimum achievable sampling error decreases, but for a fixed sample size it may or may not be possible to reach zero depending on the policy’s action probabilities.
>
> In Figures 2b and 2c of our revisions, we now include sampling error and gradient accuracy curves for an "oracle" sampling strategy that always selects the most under-sampled $(s,a)$ pair at each step. This oracle is not realizable in standard RL settings because it assumes we can reset the agent to any state at each step, but we include it to provide a better reference point for the minimum sampling error achievable in GridWorld.
>
> > In the paragraph under equation (7), it is mentioned that "We stop the PROPS update early when $D_{KL}(\pi_\theta || \pi_\phi)$ reaches a chosen threshold $\delta_{PROPS}$. Since there is also a KL regularizer between the two policies, are both of these methods necessary? It seems redundant.
>
> This is a good point for us to clarify. ​​The KL regularizer $-\lambda D_{KL}(\pi_\theta || \pi_\phi)$ in Eq. 6  limits how far the behavior policy can move from the target policy **after each gradient step.** The KL early cutoff rule $D_{KL}(\pi_\theta||\pi_\phi) > \delta_{PROPS}$ prevents the behavior policy from drifting too far **across multiple epochs of minibatch updates**, even if each step individually is small. PROPS performs up to 16 epochs of minibatch updates, so even if each step is individually “safe,” their cumulative effect can push the behavior policy too far from the target policy. The KL cutoff guards against this accumulated drift. We have clarified this point in our revisions.
>
> Figure 13 in Appendix D.3. shows that empirically, regularization is needed to ensure PROPS reduces sampling error during RL training in tasks like Hopper-v4, HalfCheetah-v4, and Walker2d-v4.
>
>
> > could the orange lines be changed to some different color? It's currently hard to tell the difference.
>
> We’ve updated the figure to use more distinguishable colors and line styles. Thank you for the suggestion!
>
> > The text refers to Fig. 6b. and mentions "5 out of 6 tasks" but the figure only shows GridWorld.
>
> We have fixed this in-text reference in our revisions. Thank you for pointing this out!
>
>
> ---
>
> Thank you again for the useful suggestions! We hope our response addresses your comments. Please let us know if there are further questions; we are happy to discuss!

---

### Author Response · Authors · 2025-12-10

We’d like to thank all reviewers for their thoughtful feedback and suggestions. We’re pleased to see that reviewers found our work well-written (`JSKr`) and believe that our insights will be useful to the RL community (`a4R9`, `JSKr`, `gaQF`). While reviewer `gaQF` noted a concern about the correctness of our claims, we have addressed these concerns in our revisions.

Before addressing specific comments, we would like to briefly reiterate our core claims and contributions, as this context helps frame our responses.

**Core Claims:** Our work shows that sampling error can cause suboptimal convergence and reduce data efficiency. The central claim we make is that PROPS reduces sampling error more than on-policy sampling and, as a result, improves data efficiency. Our experiments support this claim: Figure 5 shows that PROPS reduces sampling error during RL training and Figure 4a shows that PROPS achieves higher return with fewer environment interactions than both PPO and PPO-Buffer. Figure 4b demonstrates that PROPS shifts the distribution of final returns toward larger values, increasing the fraction of runs that achieve higher performance compared to the baselines.

We uploaded a revised paper based on reviewer comments. All changes are in blue text. Below, we summarize our revisions

1. **Additional exposition on ROS ( `a4R9`, `JSKr`):** In section 5.1, we include additional exposition on the ROS method that PROPS builds upon to make the work more self contained.
1. **Sampling error metric definition (`gaQF`):** We moved the description of our sampling error metric from the appendix to the main paper.
1. **Sampling error curves ( `JSKr`):** Sampling error figures now show results over 10 seeds, and we use more distinguishable colors and linestyles to improve readability.
1. **Oracle sampler ( `a4R9`):** Fig 2b and 2c now include sampling error and gradient accuracy curves for an "oracle" sampler that selects the most under-sampled $(s,a)$ at each step.
1. **KL Cutoff vs. KL regularization ( `a4R9`):** In footnote 4 of Section 5.2, We clarify that The KL regularizer and target KL cutoff value serve two distinct purposes: the KL regularizer in Eq. 6  limits how far the behavior policy can move from the target policy after each gradient step, while The KL early cutoff rule prevents the behavior policy from drifting too far across multiple epochs of minibatch updates, even if each step individually is small.
1. **Theory ( `JSKr`):** In the Discussion section, we clarify that our theoretical contribution focuses on the tabular setting and serves as conceptual grounding for PROPS; our result does not immediately transfer to continuous control. Our work focuses on scaling the ideas from ROS to continuous control *empirically.*
1. **Computational Cost ( `JSKr`):** We have added a cost analysis of PROPS to Appendix F.
1. **Related work (`gaQF`):** We have added discussion on Doubly Optimal (DOpt) Policy Evaluation to the related work.
1. **Infinite Horizon ( `JSKr`):** We changed our Preliminaries section to say we focus on infinite horizon MDPs.

We address reviewer-specific comments in other replies. Please let us know if our response address your comments. We are more than happy to discuss further!

---

### Decision · Action_Editor_NAuq · 2026-01-15

**Recommendation:** Accept as is

**Additional Comments:**

This manuscript addresses sampling error in on-policy reinforcement learning (RL), which arises when finite samples do not accurately represent the expected data distribution. Building on recent findings that highlight the advantages of off-policy sampling for reducing this error, the authors introduce PROPS, an on-policy policy gradient RL algorithm that incorporates an adaptive off-policy sampling method as a behavior policy to mitigate sampling error. The results demonstrate reduced error and enhanced data efficiency in MuJoCo benchmark tasks.

The reviewers agree that the work in this manuscript is well-supported and relevant to the TMLR audience. Key contributions include: (i) extending a prior approach to high-dimensional, continuous problems, (ii) improving data efficiency for on-policy policy gradient RL, and (iii) providing empirical results that support these claims.

The reviewers provided valuable comments, and authors and reviewers engaged in constructive discussions, at the end of which all reviewers confirmed that all concerns were addressed. I thank both the authors and reviewers for their active engagement throughout this process.

Based on the reviews and the outcome of the discussion phase, the manuscript is suitable and ready for publication. For the authors, please double-check that all reviewers' comments are taken into account for the final version and do a final pass for potential typos (p. 12, the sentence "Proposition 1 and previous convergence results by Zhong et al. (2022) ..." seems to be missing a period).

**Audience:**

Yes

**Audience Explanation:**

On-policy policy gradient methods are relevant and popular in modern RL; this manuscript improves data efficiency of these methods. Hence, this work is of interest to RL researchers, as well as possibly beyond (as pointed out by a reviewer).

**Claims And Evidence:**

Yes

**Claims Explanation:**

The reviewers all agree that the claims in the manuscript are supported by accurate, clear and convincing evidence, including the exposition, theoretical results, and empirical results.